# On-demand ride service platform with differentiated services

Lina Ma[1,3], Zhijie Tao[3], Qiang Wei[2,4]*

1 School of Business, Xianda College of Economics and Humanities Shanghai International Studies University, Shanghai, China, 2 School of Economics and Management, Urban Vocational College of Sichuan, Chengdu, China, 3 College of Business, Shanghai University of Finance and Economics, Shanghai, China, 4 Business School, Nankai University, Tianjin, China

* wq_research@126.com

**Data Availability Statement:** The data that support the findings of this study are available within the paper.

**Funding:** The author(s) received no specific funding for this work.

## Abstract

The rapid growth of on-demand ride service platforms has made it increasingly important for these platforms to efficiently match services by understanding driver characteristics and consumer preferences. This paper aims to investigate the pricing strategy by considering the impact of consumer preference heterogeneity and the different service types offered by drivers. The findings of this study reveal the need for the platform to strike a balance between service cost and the benefits of high-quality drivers, which can be referred to as the "cost-performance ratio". If the "cost-performance ratio" that attracts high-quality drivers is high, the platform will attract high-quality drivers or drivers of all types to participate while offering differentiated services. Otherwise, the platform will only provide services through low-quality drivers. Furthermore, the platform will also consider when to offer differentiated services based on network externalities and service quality. When the network externalities of the two types of services are similar, the platform will differentiate them based on service quality differences. Overall, considering consumer preference heterogeneity, drivers of service types, and network externalities, this paper provides guidance for platforms to make optimal decisions that enhance their service offerings and improve overall customer satisfaction.

## 1 Introduction

Platforms have revolutionized traditional business models by offering consumers convenient and efficient means to access a diverse array of products and services. For example, online shopping platforms like Amazon and Alibaba have changed the way people buy goods by offering a vast selection, competitive prices, and fast delivery. Third-party payment platforms like PayPal and Alipay have simplified online transactions and made it easier for people to make payments and transfer money securely. Sharing platforms like Uber and Airbnb have enabled individuals to monetize their assets, such as cars and spare rooms, by connecting them with people who need those services. These platform businesses have created new opportunities for entrepreneurs and individuals to generate income and have empowered consumers with more choices and convenience [1–4]. The platform sharing economy is still evolving, and

**Competing interests:** The authors have declared that no competing interests exist.

its impact on various industries and society as a whole is still being studied. It is crucial for policymakers, businesses, and individuals to understand and adapt to this new economic model to ensure its benefits are maximized while addressing any potential challenges that may arise. The platform also handles the payment process, ensuring a secure transaction for both parties involved [5, 6]. The platform's operation is crucial for the success of the on-demand economy. It facilitates the connection between consumers and service providers, sets prices, ensures trust and safety, and provides the necessary technology for seamless transactions [7]. It typically includes mobile apps or websites that allow consumers to easily request rides and drivers to accept or decline requests. The platform is responsible for matching drivers with consumers based on various factors, such as location and availability [8].

As a type of two-sided market, an on-demand ride service platform acts as an intermediary between drivers and consumers. Its distinguishing feature is the cross-network externality, where the utility of one user is influenced by the size of the other group [9]. Therefore, the initial goal of an on-demand service platform is to ensure a sufficient number of participants, as the cross-network externality is a distinguishing feature of two-sided markets. On one hand, when there is a high number of participating drivers, consumers are more likely to join the platform due to shorter waiting times. On the other hand, a high number of consumers increases driver participation in the platform due to the availability of numerous orders. Consequently, a mutually agreeable wage or price is established for trading purposes. The platform, acting as an intermediary, receives a percentage of the profits from each order, in accordance with the agreed arrangement for all parties involved.

In previous studies on platform differentiation, some scholars have linked differentiation with network effects, suggesting that in industries with network effects, network size is a more important factor of competition than quality [10]. Later literature has defined service quality as network externalities [11]. With the development of two-sided markets, both sides have heterogeneous preferences and features. Consumers' preferences influence their participation in decision-making processes [12]. Firstly, consumers have varying preferences for service quality. Some are willing to pay a higher price for high-quality service, while others are willing to accept lower service quality at a lower price as long as the waiting time is appropriate. Additionally, drivers themselves have different levels of service quality, including the type of vehicles and diverse service levels provided by drivers [13]. With the booming of on-demand service platforms, it is indeed necessary for platforms to provide different experiences in order to carve out their own niche. Additionally, platforms need to cater to consumers with different service preferences by offering various types of services. In reality, platforms like Didi provide differentiated service types such as express, private car, and hitch. Each service type corresponds to a different type of driver, and consumers who choose these services generally have similar preferences. By differentiating services according to driver types, the platform can maximize the utility and benefits for all participants to a greater extent. For example, the platform can match high-quality services with high-quality preference consumers, thereby implementing price discrimination. So it becomes crucial for the platform to identify the different features or requirements of both sides (consumers and drivers) and find ways to achieve more accurate matching.

Based on consumers' preferences and the varying levels of service provided by drivers, the platform aims to achieve more precise matching between consumers and providers [14]. One approach is to allow all drivers to offer a hybrid service, which can generate a pooling effect that is greater compared to when they provide separate services. The platform can also make different service decisions based on various conditions. For instance, if the number of consumers is not very large, the platform may consider having only a portion of the service providers offer their services to save costs. On the other hand, if the number of consumers is large, the

service will be provided by all drivers, allowing the platform to benefit from high network externalities. To determine whether service differentiation is beneficial for platforms and when to implement it, as well as to identify the optimal type of driver to provide service under different conditions, this paper takes into account the characteristics of network externalities in the context of on-demand service platforms. It also considers the heterogeneity of consumers' service preferences on the demand side and the different service types of drivers on the supply side, aiming to explore the optimal pricing decision for the platform.

The remaining sections of the paper are organized as follows. Section 2 provides a review of the relevant literature. Section 3 and 4 present the analysis of the models with differentiated and non-differentiated service types respectively. In section 5, we present our numerical analysis. Finally, section 6 concludes the paper.

## 2 Literature review

In this section, we provide a summary of the literature on two aspects: (1) research on two-sided markets, and (2) the impacts of service quality on decision making. The first aspect examines the relationship between previous research on two-sided markets and our own work. The second aspect reviews existing studies on service quality, some of which support our key assumptions and model development.

### 2.1 Researches on two-sided market

The popularity of two-sided markets in real life has drawn the attention of scholars to the related issues in this field. Instead of directly providing products or services, platforms act as "intermediary" connecting both sides of the market. The on-demand service platform, which is the focus of this paper, is just one example of such platforms, with similar forms found in rental markets [15], software development markets, and so on. The primary research areas in this field include the pricing behavior of platform owners [16], the matching mechanisms employed by the platform [17], and the effects of new platforms entering the market on existing ones [18].

One prominent characteristic of two-sided markets is the presence of cross-network externalities. This means that the participation of service providers on the platform can impact the utility of demanders, thus influencing their decision to participate. In a two-sided market, the benefits of one party are closely tied to the scale of participation by the other party on the platform [9]. Users on both sides are considered valuable internal resources for the platform, and the initial user base is crucial for maintaining a competitive advantage and influencing long-term competition [19]. In a two-sided market, users on both sides derive utility or income by engaging in transactions on the platform. The impact of network externalities on pricing in two-sided markets has been studied extensively, including cross-group network externalities and intra-group network externalities [20, 21]. Many scholars argue that two-sided platforms exhibit the typical characteristics of cross-network externalities. This paper recognizes this feature and incorporates it into the utility function of service demanders.

The popularity of on-demand ride service platforms has attracted significant attention from scholars studying pricing decisions in this context. Some papers focus on the issue of price and wage incentives to effectively coordinate the supply and demand on these platforms [22–24]. Guda and Subramanian [25] have studied the impact of surging price on driver positioning in a two-stage model. They find that drivers may not trust the demand prediction, so price changes can increase the credibility of the prediction. However, increasing prices for high-demand location-based areas may not always be the most effective strategy, as it can suppress demand growth or even drive drivers away from those areas. Similarly, there have been

matching studies that show how platform incentives for agents can influence the number of agents [26]. In this paper, the optimal decision of the platform is determined by considering the service quality of drivers and balancing the supply and demand. The model is built upon the assumptions of previous studies.

## 2.2 Service quality on decision making

Currently, research on the service quality of platforms is mostly focused on the impact of waiting time during travel, as long waiting times often reduce consumers' service experience [27]. Considering consumers' sensitivity or impatience towards waiting time, it is important to develop price and wage strategies to maximize profits under the assumption that consumers have low tolerance for waiting time [28]. Some consider how to enhance the matching efficiency of platforms. They thought platforms can improve service quality in the following aspects. Zhou et al. [29] investigated the optimal pricing decisions of service enterprises with the presence of two groups of consumers having different valuation and service sensitivity. Ni et al. [30] studied the optimal pricing and service speed of a platform with two types of consumers. The study found that under the goal of maximizing profits, it is not always optimal to serve all consumers. The waiting time acts as a determinant of demand, indicating the impact of service quality [11].

In fact, consumers have different preferences and priorities, so differences in service quality need to be considered in some problems. The preferences of consumers for platform service quality play a crucial role in determining the platform's optimal decision strategy [31]. And service network externalities are often considered as part of service quality [32–34]. This paper incorporates the heterogeneity of consumers' service preferences in our model. Similar to the problem in the paper of Zhong et al. [35], we studied the influence of each parameter on the optimal decision of the platform from the perspective of distinguishing and not distinguishing service quality. Similar to the problem addressed in Zhong et al.'s paper, our study examines the impact of various parameters on the optimal decision-making of the platform, focusing on the distinction and non-distinction of service quality. While they concluded that serving all consumers is not always optimal, our article takes a different approach by defining service quality as the consumer's sensitivity to congestion levels. The consumer's utility includes intrinsic evaluation, price, and waiting time, which serves as a proxy for service quality. By comparing the results, they illustrate the conditions under which differentiated or undifferentiated services are applicable and conclude that blindly serving all consumers is not the optimal strategy for the platform. Building upon these findings, our paper extends the concept of service quality by incorporating the consumer's experience as a measure, in addition to considering waiting time. Furthermore, we also take into account the cross-network externalities. Recognizing the heterogeneity in service quality among consumers, we explore the service provider's role and examine the optimal decision-making process while considering the service quality provided by the driver as parameters in our model.

Existing literature on the service quality of on-demand ride platforms mainly focuses on exploring the impact of service differences on both sides of the platform, or considers service preference as fixed parameter. However, considering the characteristics of cross-network externalities in a two-sided market, this paper takes into account the service quality types of drivers and the heterogeneous preferences of consumers for driver service quality. Our study aims to investigate the optimal pricing strategy for the platform in light of these factors.

Considering all the above research on on-demand service platform, we can fill the following gaps in literature: (1)In this paper, the service provider's service quality is taken into consideration. Combined with the feature of two-sided market: cross network externalities, we

considers the consumer's heterogeneity preference, service quality to explore how consumers integrated all factors to make decisions. Backward induction method is used to determine the platform optimization strategy. (2)This study takes into account the service quality of the service providers, as well as the cross network externalities inherent in a two-sided market. By considering the heterogeneity in consumer preferences and the impact of service quality on their decision-making process, we aim to understand how consumers integrate all these factors to make their choices. In the context of this study, the platform initially enters the market by offering a single type of service without differentiating between providers. All providers pool together to offer this service to consumers. The platform's decision variable is a single set of price and wage. This paper is the first to compare platform decisions with different parameters and identify the conditions under which the platform should provide a single service or opt for differentiated services. By analyzing the impact of various factors, such as provider service levels and retention costs, the study determines the optimal strategy for the platform to maximize its profits while satisfying consumer preferences. Overall, by addressing these gaps, our research contributes to a more comprehensive understanding of service quality and pricing strategies in on-demand ride platforms. We provide valuable insights for platform decision-makers to optimize their strategies and enhance the overall service quality and customer satisfaction.

## 3 Problem description and basic model

Usually, when a platform enters the market, it only provides one type of service without differentiating the service providers. All drivers are mixed together to offer services to consumers, and the platform's decision variables are limited to price and wage. However, as the platform develops, it starts considering the service preferences of different consumers and the varying service levels of drivers. For instance, consumers who prioritize price over service quality will be more price-sensitive and pay less attention to the quality of services. In this case, the platform monetizes its operations by attracting a large consumer base through low prices. On the other hand, there are consumers who prioritize service level over price. They are willing to pay a higher price for a higher level of service. To cater to these consumers, the platform invests significantly in achieving a high level of service, thus incurring higher costs. The platform then generates profits by charging a higher price for these high-quality services. In this context, there is currently limited research on the optimal strategy for launching different types of services.

Therefore, the platform has found that drivers have varying service levels and retention costs as the platform develops. Naturally, drivers who provide high-quality services tend to have higher retention costs. Consequently, the platform must consider whether to pursue greater profits through service differentiation.

Based on the theory of two-sided markets, the platform has two types of users: service demanders and service providers. In this market, it is assumed that there are two types of service providers: low quality and high quality. For example, in the on-demand service industry, this can be represented by different levels of car types or the drivers' previous customer satisfaction ratings. The platform categorizes service providers into high-type and low-type based on certain characteristics, denoted as H and L type service providers. The corresponding service quality is represented as $q_H$ for high-type service providers and $q_L$ for low-type service providers. Similarly, the retention costs are denoted as $r_H$ for high-type service providers and $r_L$ for low-type service providers. The potential number of high-type service providers is denoted as $N_H$, and the potential number of low-type service providers is denoted as $N_L$. It is assumed that $q_H > q_L$, $r_H > r_L$, and $N_H < N_L$. The number of potential service providers refers

to the number of registered service providers on the platform. It represents the total pool of service providers available for users to choose from.

After registering on the platform, the service provider (in this case, the driver) will evaluate the salary or income offered by the platform. If the income exceeds the retention cost, the provider will decide to provide services for the platform. However, if the income is less than the retention cost, the provider will not provide services. On the other hand, the service demanders have heterogeneous preferences for the service quality of the providers. Each service demander decides whether to participate or use the platform's services based on their own utility. This means that service demanders will consider factors such as the service quality, price, convenience, and other aspects to determine if using the platform is beneficial to them.

The event sequence is as follows:

Service providers register on the platform. Service demanders search for services on the platform. The platform analyzes the number of potential providers and the demand for services. The platform sets a unit price for service demanders and a wage for service providers based on this analysis. Service demanders evaluate the unit price and their own utility to determine whether to participate in the platform. If service demanders find the unit price and the platform beneficial, they decide to participate and use the services. Service providers evaluate the offered wage and their own income needs. If the offered wage exceeds the income needs of service providers, they decide to provide services for the platform. Participating service demanders and registered service providers connect on the platform.

We assume that the platform has access to information about the service types of all the drivers, so it can categorize service providers into two types: H and L. Additionally, the platform can also classify service providers based on observable factors such as vehicle models or driver service evaluations. Instead of distinguishing service types, the platform can choose two types of service providers and mix them together to provide services to the demanders. In this scenario, the network externality is maximized, and the demanders can only observe and choose from one type of service. As a result, the platform's decision variables are reduced to a set of decision variables $(P, w)$. This case is known as mixed case, referred to as case M; If the service types are distinguished, with H type representing a high-quality level and L type representing a low-quality level, then demanders can observe and choose from two different types of services. In this case, the platform has two sets of decision variables: $(P_H, w_H)$ for the high-quality service type and $(P_L, w_L)$ for the low-quality service type. This scenario is known as the separate case, referred to as case S.

In this case, the consumers are the service demanders, and the drivers are the service providers. The drivers who provide a high-quality level of service will be referred to as H-type drivers, while those who provide a low-quality level of service will be referred to as L-type drivers. In accordance with the sequence of events, backward induction is used to solve question.

The main symbols and descriptions are shown in Table 1.

Next, let's take a look at the case S, which distinguishes quality of service types.

In the case S, we introduce the concept of distinguishing quality of service types for the platform and the drivers.

## 3.1 Differentiated service types

In many studies, the utility function for both users in a two-sided market is often assumed to be the same. However, considering the unique characteristics of users in on-demand ride service platform, this paper takes into account different factors for consumers and drivers participation. Consumers take the external effects, prices, and service quality of drivers into consideration, whereas drivers prioritize their income and the efficiency of receiving orders.

**Table 1. List of symbols and descriptions in the model.**

| Symbols | Descriptions |
| --- | --- |
| $P_i$ | The unit price of $i$ type driver service for consumer |
| $w_i$ | Unit wage for $i$ type services driver |
| $P_j$ | Unit price for consumer in $j$ case |
| $w_j$ | Unit wage for drivers in the $j$ case |
| $n_{ci}$ | The number of consumers served per unit time by $i$ type drivers |
| $n_{di}$ | The number of $i$ type drivers participating per unit time |
| $r_i$ | Retention costs of $i$ type drivers |
| $q_i$ | Service quality of $i$ type drivers |
| $N_i$ | The potential total number of $i$ type drivers |
| $\alpha_j$ | Cross-networks externalities in the $j$ case |
| $\xi$ | Consumer preference of service quality |
| $i$ | $i = L, H$ refer to two types of drivers respectively |
| $j$ | $j = L, H, M$ refer to different cases |

To model the utility function for drivers, this paper draws on the form used by Zhong et al. (2020). It considers that drivers make their participation decision based on their income, which is influenced by the platform salary, the efficiency of receiving orders, and their own retention costs. This more comprehensive utility function captures the various factors that influence driver participation.

The platform classifying drivers into high quality (H-type) or low quality (L-type) based on certain characteristics is a common practice in many on-demand service platforms. This classification helps the platform differentiate between drivers and provide different levels of service to customers based on their preferences and willingness to pay. The unit salary given by the platform is $w_H$ and $w_L$ respectively. Driver's revenue function is $R_{di} = \frac{w_i \cdot n_{ci}}{n_{di}} - r_i$, $i = L, H$. $n_{ci}$ is the number of consumers choosing $i$ type, $n_{di}$ is the number of $i$ type drivers, assume that every consumer create only one service demand, so, $\frac{n_{ci}}{n_{di}}$ is the order volume of each driver. If $R_{di} \geq 0$, the driver will participate. Assuming that the two types of drivers are homogeneous, the providers either do not participate or choose to participate, so $n_{d_H} = N_H$, $n_{d_L} = N_L$. It is optimal for platform to provide wage that satisfies $R_{di} = 0$, namely $\frac{w_i \cdot n_{ci}}{N_i} = r_i$. Therefore, the salaries given by the platform are:

$$w_H = \frac{r_H \cdot N_H}{n_{cH}}$$

$$w_L = \frac{r_L \cdot N_L}{n_{cL}}$$

In addition to consumers' preference for services, this paper also considers the cross-network externalities that influence consumer utility. When consumers observe the prices $P_H$ and $P_L$ offered by the platform for the two types of services, they choose either the low or high quality service based on their utility.

Consumer utility is affected by driver service quality, cross-network externalities, and price. The preference for driver service quality is denoted as $\xi$, $\xi \sim U[0, 1]$. $\alpha_i$ represents the external utility of the network when the number of potential providers is $N_i$. The higher the number of potential providers is, the higher the network externality will be brought to the demanders,

because $N_L > N_H$, so we assume $\alpha_L > \alpha_H$. Then the utility function of consumers can be expressed as:

$$U_i = \xi \cdot q_i^e + \alpha_i - P_i$$

Where $i = H, L$.

When the utility of choosing either H type or L type service is greater than or equal to 0, consumers will participate. In this case, $\xi \cdot q_L + \alpha_L - p_L \geq 0$, $\xi \cdot q_H + \alpha_H - p_H \geq 0$, then the two critical points for consumers to choose to participate are $\xi_L = \frac{P_L - \alpha_L}{q_L}$, $\xi_H = \frac{P_H - \alpha_H}{q_H}$. According to $U_L = U_H$, we have $\xi_{LH} = \frac{P_H - \alpha_H - (P_L - \alpha_L)}{q_H - q_L}$. Assuming that $\xi_H \geq \xi_L$, then $\xi_{LH} \geq \xi_H$. The distribution of consumer choice can be seen in Fig 1.

Based on the information provided, consumer choice depends on a parameter $\xi$. When $\xi$ falls within the range $[\xi_L, \xi_{LH}]$, the consumer will choose the L type service. When $\xi$ falls within the range $[\xi_{LH}, 1]$, the consumer will choose the H type service. The specific values of $\xi_L$ and $\xi_{LH}$ would determine the threshold at which consumers switch from choosing L type to H type service.

$$
\begin{aligned}
n_{cL} &= Prob\{\xi_L \leq \xi \leq \xi_{LH}\} \\
&= \frac{P_H - \alpha_H - (P_L - \alpha_L)}{q_H - q_L} - \frac{P_L - \alpha_L}{q_L}
\end{aligned}
$$

$$
\begin{aligned}
n_{cH} &= Prob\{\xi_{LH} \leq \xi \leq 1\} \\
&= 1 - \frac{P_H - \alpha_H - (P_L - \alpha_L)}{q_H - q_L}
\end{aligned}
$$

From above, we can use the following expressions for $P_L$ and $P_H$:

$$P_L = q_L \cdot (1 - n_{cL} - n_{cH}) + \alpha_L$$

$$P_H = q_H \cdot (1 - n_{cH}) - q_L \cdot n_{cL} + \alpha_H$$

We assume that the platform can cover all market demands and that operating costs are not taken into account, the optimization problem can be simplified as $\pi = n_{cH} \cdot (P_H - w_H) + n_{cL} \cdot$

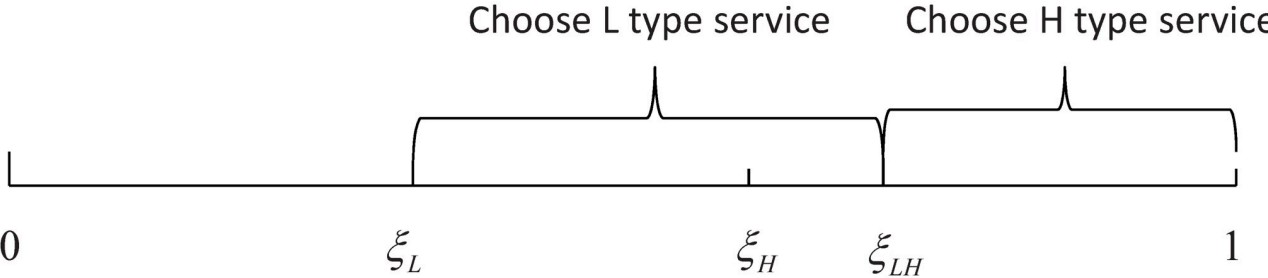

**Fig 1. The distribution of consumer choice.**

$(P_L - w_L)$, so:

$$max_{n_{cL}, n_{cH}} \pi = q_H \cdot n_{cH} - (q_H - q_L) \cdot n_{cH}^2$$
$$-q_L \cdot (n_{cL} + n_{cH})^2 + \alpha_H \cdot n_{cH} - r_H \cdot N_H + q_L \cdot n_{cL}$$
$$+\alpha_L \cdot n_{cL} - r_L \cdot N_L$$
$$s.t. \quad n_{cL} + n_{cH} \in [0, 1],$$
$$n_{cH} \in [0, N_H],$$
$$n_{cL} \in [0, N_L],$$

According to the first-order condition for profit maximization, we have $\frac{\partial \pi}{\partial n_{cH}} = 0 \frac{\partial \pi}{\partial n_{cL}} = 0$, and,

$$n_{cH}^* = \frac{1}{2} - \frac{\alpha_L - \alpha_H}{2(q_H - q_L)}$$

$$n_{cL}^* = \frac{1}{2} \left[ \frac{\alpha_L}{q_L} + \frac{\alpha_L - \alpha_H}{(q_H - q_L)} \right]$$

$$n_c = n_{cH}^* + n_{cL}^* = \frac{1}{2} \cdot \left( 1 + \frac{\alpha_L}{q_L} \right)$$

so

$$P_L^* = \frac{q_L + \alpha_L}{2}$$

$$P_H^* = \frac{q_H + \alpha_H}{2}$$

The monotonicity of decision variables on each parameter can be obtained:

**Proposition 3.1** *Driver's wage $w_H$ decreases with $q_H$ and $\alpha_H$, increases with $q_L$ and $\alpha_L$; $w_L$ decreases with $q_L$ and $\alpha_L$, increases with $q_H$ and $\alpha_H$.*

The wage for drivers $w_H$ decreases with $q_H$ and $\alpha_H$, while it increases with $q_L$ and $\alpha_L$. On the other hand, the $w_L$ decreases with $q_L$ and $\alpha_L$, but increases with $q_H$ and $\alpha_H$. The Price for consumers $P_H$ increases with $q_H$ and $\alpha_H$; $P_L$ increases with $q_L$ and $\alpha_L$.

If the service quality of drivers improves or external factors have a positive impact, the number of consumers opting for this type of service is likely to increase, leading to a corresponding decrease in the number of consumers choosing another type of service. Additionally, the relationship between price, salary, and the number of consumers adheres to the principles of market demand and price elasticity.

## 3.2 Non-differentiated service type

In the development process of a platform, it is common for platforms to initially provide a single type of product or service without considering consumer service preferences. In this context, we propose the idea of pooling both high quality (H) and low quality (L) drivers together to provide services. This approach aligns with the concept of a demand-oriented two-sided platform. The two types of drivers are homogeneous, with high-quality drivers providing a service level of $q_H$ and low-quality drivers offering a service level of $q_L$. On the other side of the platform are consumers, whose preference for service quality is represented by $\xi$, $\xi \sim U[0, 1]$,

The platform only provides one service, which involves pooling together the two types of drivers to provide services for consumers. Consequently, the platform has only one set of decision variables, represented by $(P, w)$.

Firstly, the revenue of the drivers is taken into consideration. If the two types of drivers are homogeneous, then when the revenue falls below a certain threshold $r_L$, no driver will choose to participate. When the revenue is greater than or equal to $r_L$ but less than $r_H$, all drivers of type L participate. However, when the revenue exceeds $r_H$, all drivers, including both L and H types, participate.

**Lemma 3.1** *Assuming that high-quality (H) and low-quality (L) drivers are respectively homogeneous, the number of driver participation is*:

$$n_d^* = \begin{cases} 0 & \dfrac{wn_c}{n_d} < r_L \\[2mm] N_L & r_L \leq \dfrac{wn_c}{n_d} < r_H \\[2mm] N_L + N_H & r_H \leq \dfrac{wn_c}{n_d} \end{cases}$$

Therefore, by considering the revenue thresholds for driver participation, we can determine the number of participating drivers, the expected service level for consumers, and the unit wage provided by the platform to drivers. These values can be organized and presented in Table 2.

In other words, there are two different scenarios for driver participation in the platform. In case L, the platform aims to attract only L-type drivers, while in case M, it aims to attract all drivers, including both L and H types (Mixed-Case). Each scenario is associated with a specific price and wage offered by the platform.

In addition, if the platform does not distinguish between driver service types and only provides one type of service for consumers, it can also choose to exclusively select H-type drivers to provide the service. This scenario is referred to as case H. In this case, the selected H-type driver can provide a service quality denoted as $q_H$. The service quality $q_H$ is equivalent to the consumer's expectation of service quality $q^e$. The retained cost for the H-type driver is denoted as $r_H$, and the total number of potential H-type drivers available is denoted as $N_H$. The platform in this scenario has only one set of price and wage as its decision variables.

Similarly, the H-type drivers in this scenario also have only one set of wage determined by the platform as their compensatio, $w = \frac{r_H N_H}{n_c}$.

The utility function of consumers consists of three components: service quality preference, cross-network external utility, and price.

$$U_j = \xi \cdot q_j^e + \alpha_j - P_j$$

Where $j = L, H, M$.

**Table 2. Equilibrium of driver's participation under a service type.**

| Driver's revenue | Number of drivers | [c]Consumers' expected service quality | Driver's wage |
|---|---|---|---|
| $0 < \frac{wn_c}{n_d} < r_L$ | 0 | —— | —— |
| $r_L \leq \frac{wn_c}{n_d} < r_H$ | $N_L$ | $q_L$ | $\frac{r_L N_L}{n_c}$ |
| $r_H \leq \frac{wn_c}{n_d}$ | $N_L + N_H$ | $\frac{q_L N_L + q_H N_H}{N_L + N_H}$ | $\frac{r_H (N_L + N_H)}{n_c}$ |

Previously, $\alpha_L$ and $\alpha_H$ represented the external utility of the network effects when the number of potential drivers in the L-type and H-type categories are $N_L$ and $N_H$, respectively. $\alpha_L$ remains the same as before, representing the cross-network effect when the number of potential drivers is $N_L$. $\alpha_M$ represents the network effect when the number of potential drivers is $N_L + N_H$. As the number of potential drivers increases, the network effect on demanders also increases. Therefore, when $N_L + N_H > N_L > N_H$, we have $\alpha_M > \alpha_L > \alpha_H$.

Consumers will participate as long as their utility $U_j \geq 0$. The critical point at which consumers decide to participate is:

$\xi_0 = \frac{P_j - \alpha_j}{q_j^e}$.

$$n_c = Prob\{\xi_0 \leq \xi \leq 1\} = 1 - \frac{P_j - \alpha_j}{q_j^e}$$

$$P_j = q_j^e \cdot (1 - n_c) + \alpha_j$$

It is also assumed that the platform can meet all market demands, so the platform's optimization problem is as follows:

$$max_{n_c} \pi = q_j^e \cdot (n_c - n_c^2) + \alpha_j \cdot n_c - r_j \cdot n_d$$
$$s.t. \quad n_c \in [0, 1]$$

According to the first order condition for profit maximization, we have $\frac{\partial \pi}{\partial n_c} = 0$, so the optimal number of consumer participating is,

According to the first-order condition for profit maximization, we have $\frac{\partial \pi}{\partial n_c} = 0$. This condition implies that the optimal number of consumers participating is:

$$n_c^* = \frac{1}{2} \cdot \left(1 + \frac{\alpha_j}{q_j^e}\right)$$

And the optimal price is,

$$P_j^* = \frac{q_j^e + \alpha_j}{2}$$

## 4 Comparative analysis

According to the first-order condition of profit maximization, the optimal solution and optimal profit in the four cases are shown in Table 3.

In the above table, it is stated that the optimal solution for $q^e$ is given by $\frac{q_L N_L + q_H N_H}{N_L + N_H}$. Additionally, it is mentioned that the optimal profit in strategy S remains the same regardless of the form used.

In this model, consumer utility is influenced by various parameters, including external effects, service quality, and price. While we consider the cross network effect, we assume that the preference for service quality has a greater impact compared to external effects. This assumption is made because the primary focus of this paper is to examine heterogeneity in consumer service quality preferences. Specifically, we assume that $\alpha_j \leq q_j^e$, which implies that $\frac{\alpha_j}{q_j^e} \leq 1$. Consequently, we can deduce that $n_c^* \leq 1$. Next, we can compare the optimal profits of the platform under different cases to determine the conditions under which it is optimal for

**Table 3. The optimal solution for the four cases.**

| Strategy | Optimal price | Optimal consumer number | Optimal profit |
|---|---|---|---|
| Strategy H | $\frac{q_H+\alpha_H}{2}$ | $\frac{1+\alpha_H/q_H}{2}$ | $\frac{(q_H+\alpha_H)^2}{4q_H} - r_H N_H$ |
| Strategy L | $\frac{q_L+\alpha_L}{2}$ | $\frac{1+\alpha_L/q_L}{2}$ | $\frac{(q_L+\alpha_L)^2}{4q_L} - r_L N_L$ |
| Strategy M | $\frac{q^e+\alpha_M}{2}$ | $\frac{1+\alpha_M/q^e}{2}$ | $\frac{(q^e+\alpha_M)^2}{4q^e} - r_H(N_L + N_H)$ |
| Strategy S | $P_H^* = \frac{q_H+\alpha_H}{2}$ $P_L^* = \frac{q_L+\alpha_L}{2}$ | $n_{cH}^* = \left(1 - \frac{\alpha_L-\alpha_H}{q_H-q_L}\right)/2$ $n_{cL}^* = \left(\frac{\alpha_L}{q_L} + \frac{\alpha_L-\alpha_H}{q_H-q_L}\right)/2$ | $(1)\frac{(q_H+\alpha_H)^2}{4q_H} - r_H N_H - \frac{\alpha_H^2}{4q_H} + \frac{\alpha_L^2}{4q_L} + \frac{(\alpha_H-\alpha_L)^2}{4(q_H-q_L)} - r_L N_L$ $(2)\frac{(q_L+\alpha_L)^2}{4q_L} - r_L N_L + \frac{[q_H-q_L-(\alpha_L-\alpha_H)]^2}{4(q_H-q_L)} - r_H N_H$ |

the platform to choose a specific type of driver to provide services. By analyzing the profitability in each case, we can draw conclusions about the optimal driver selection for the platform.

**Proposition 4.1** *let $\alpha$ represent a certain value, $\Delta q$, $\Delta r$ represent incremental changes in quantities and costs. If $\alpha_M$ is under a certain value ($\alpha_M \leq \alpha$), the following conditions apply:*

*(1)When $r_H - r_L \leq \Delta r$, the optimal choice is $\pi_H$.*

*(2)When $r_H - r_L \geq \Delta r$ and $q_H - q_L \leq \Delta q$, the optimal choice is $\pi_L$.*

*(3) When $r_H - r_L > \Delta r$ and $q_H - q_L > \Delta q$, the optimal choice is $\pi_H$.*

**Proof.** If $\alpha_M$, $\alpha_L$, and $\alpha_H$ are small enough to have a negligible effect on profit, we can consider them insignificantly impacting the profit function. So $o(1)$ represents the influence of external effects on the profit function, which can also be deemed negligible. Then,

$$\pi_H = \frac{q_H}{4} - r_H N_H + o(1)$$

$$\pi_S = \frac{q_H}{4} - r_H N_H - r_L N_L + o(1)$$

$$\pi_L = \frac{q_L}{4} - r_L N_L + o(1)$$

$$\pi_M = \frac{q^e}{4} - r_H(N_L + N_H) + o(1)$$

From the analysis above, we can conclude that $\pi_H > \pi_S$ and $\pi_H > \pi_M$. Thus, it is necessary to compare $\pi_H$ and $\pi_L$. Moreover, since $N_L > N_H$, we observe that when $r_H$ is small (close to $r_L$), $\pi_H > \pi_L$. As $r_H$ increases, if $r_H N_H < r_L N_L$, then $\pi_H > \pi_L$. However, when $r_H$ becomes sufficiently large and the difference with $r_L$ is significant, if $q_H$ is small (close to $q_L$), we find that $\pi_H < \pi_L$.

Since $\pi_H - \pi_L = \frac{q_H-q_L}{4} + r_L N_L - r_H N_H + o(1)$, it can be observed that the higher the value of $r_H$, the higher the value of $q_H$ at the critical point where $\pi_H = \pi_L$.

In this case, Fig 2 illustrates the regions where the optimal profit is located.:

It is indicated that when the external effects are small, the platform will choose $H$ drivers to participate in providing services and achieve the highest profit if the unit retained cost of $H$ drivers is relatively low or close to that of $L$ drivers. However, if the unit retained cost of $H$ drivers is higher than that of $L$ drivers and there is little difference in service quality between $H$ and $L$ drivers, the platform will choose $L$ drivers to provide services. On the other hand, if $q_H$ is much higher than $q_L$, it is optimal for the platform to choose $H$ drivers to provide services.

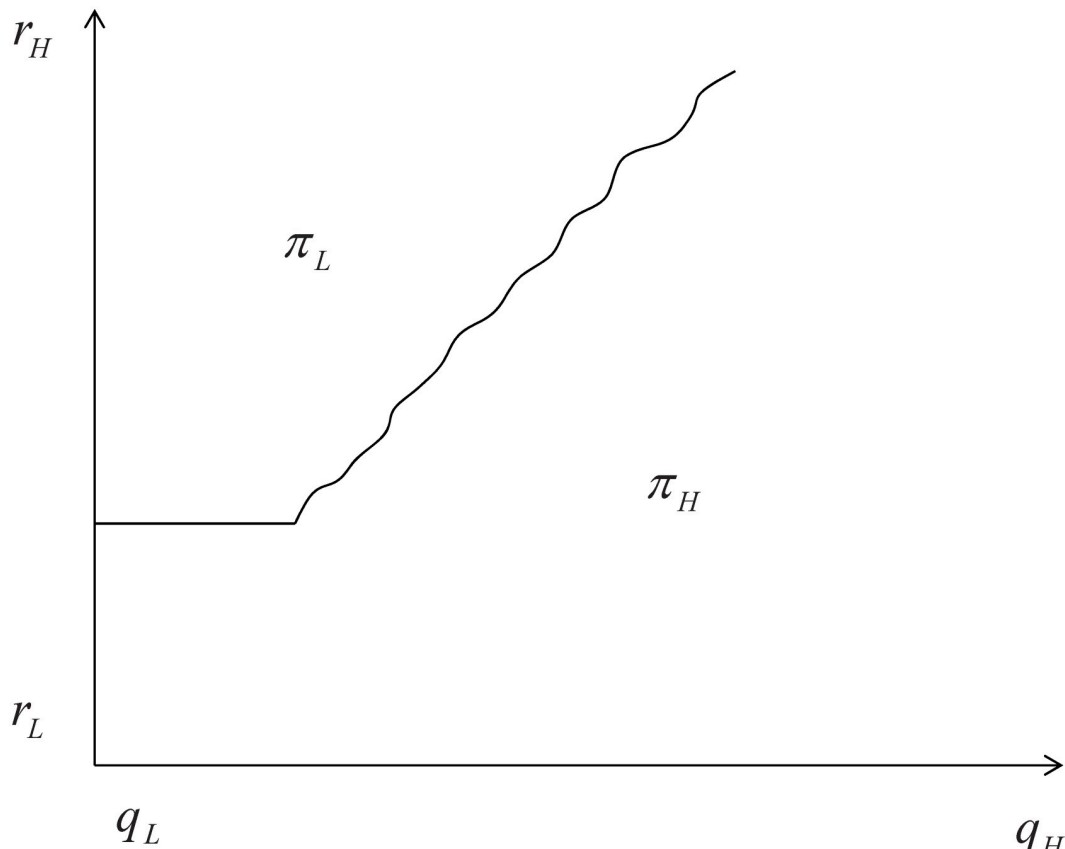

**Fig 2. The optimal profit with small externalities.**

In other words, when the external effects are small, the platform will take into account both the cost and service quality. If there is little difference in cost between the two types of drivers, the platform will prioritize high-quality service providers. This is because, when there is little difference in cost, choosing high-quality service providers can help attract more consumers and ultimately lead to higher profits for the platform. When there is a significant cost difference between the two types of drivers, the platform needs to consider the extent to which high-quality service can be provided by the high-quality service provider. If the difference in service quality between the high-quality and low-quality providers is not significant, it may not be worth bearing the high cost of the high-quality provider. In such a case, the platform may choose the low-quality provider. However, if the service quality provided by the high-quality provider is significantly higher than that of the low-quality provider, the platform has an incentive to incur the higher cost and select the high-quality drivers to participate in order to provide a better experience for consumers. Ultimately, the platform's decision will depend on the trade-off between cost and service quality.

When the external effect becomes more significant, it necessitates a more thorough analysis. Firstly, let's consider the scenario where the retained cost of L-type units is relatively low.

**Proposition 4.2** *let $\bar{\alpha}$ represent a certain value, $r$ denote a specific threshold, and $\Delta q_1$, $\Delta q_2$, $\Delta r_1$, and $\Delta r_2$ represent incremental changes in quantities and costs. It is assumed that $\alpha_H$ is greater than or equal to $\bar{\alpha}$, and if $r_L$ is less than or equal to $r$, the following conditions apply:*

*(1) When $r_H - r_L \leq \Delta\, r_1$, if $q_H - q_L \leq \Delta\, q_1$, the optimal choice is $\pi_M$. However, $q_H - q_L > \Delta q_1$, the optimal choice is $\pi_S$.*

*(2) When $\Delta r_1 < r_H - r_L \leq \Delta\, r_2$, the optimal choice is $\pi_S$.*

*(3) When $r_H - r_L > \Delta r_2$, if $q_H - q_L \leq \Delta\, q_2$, the optimal choice is $\pi_L$. However, if $q_H - q_L > \Delta q_2$, the optimal choice is $\pi_S$.*

**Proof.** When the externality effect becomes larger,

$$\pi_S - \pi_H = \frac{(q_L \alpha_H - q_H \alpha_L)^2}{4 q_L q_H (q_H - q_L)} - r_L N_L$$

$$\pi_S - \pi_L = \frac{[(q_H + \alpha_H) - (q_L + \alpha_L)]^2}{4(q_H - q_L)} - r_H N_H$$

Since the value of $r_L$ is small, based on the formula, there exists a value of $r_L$ that satisfies $r_L \leq r$, resulting in $\pi_S > \pi_H$. Furthermore, when $r_H$ is small, let's consider the extreme case where $r_L = r_H = 0$. In this situation, we can observe that $\pi_S > \pi_L$, prompting a comparison between $\pi_S$ and $\pi_M$.

$\pi_M$ achieves its optimal value when there is a small difference between $q_H$ and $q_L$, or when $\alpha_M$ is large.

Due to the increasing cost of case M with respect to $r_H$, $\pi_S$ becomes the optimal choice. The total quantity, $n_c$, remains constant in both case $L$ and case $S$. However, when $r_H$ reaches a sufficiently large value, the cost of case $S$ surpasses that of case $L$. Specifically, in case $S$, $P_H = \frac{q_H + \alpha_H}{2}$ and $P_L = \frac{q_L + \alpha_L}{2}$, while in case $L$, $P = \frac{q_L + \alpha_L}{2}$.

If the value of $q_H$ is close to that of $q_L$, then the revenue in case $S$ will not be substantially higher than in case $L$, and in fact it may even be lower. This is because $\alpha_H$ is smaller than $\alpha_L$. However, if $q_H$ is significantly larger than $q_L$, then the profit of case $S$ will be greater than that of case $L$.

Fig 3 illustrates the regions where the optimal profit is located in this case.:

When the mixed external effects are significant, the unit retained cost of high-quality drivers is comparatively low, and the expected service quality of consumers is close to $q_L$, the platform will opt to mix the two types of drivers without differentiating service quality. This observation aligns with our numerical findings.

In this case, as the retained cost of high-quality drivers increases, the cost of mixing the two types of drivers also rises, making differentiated service $S$ more optimal. However, once the retained cost of high-quality drivers surpasses a certain threshold, the participation cost for these drivers becomes prohibitively high, leading to a trade-off between service quality and cost. If both types of drivers offer similar service quality, the platform will only attract low-quality drivers. Conversely, if the high-quality level significantly surpasses the low-quality level, the platform will select both types of drivers and distinguish between service types, implementing strategy $S$ to achieve differential pricing.

When the retained cost of $L$ type drivers is low and the external effect is significant, strategy $M$ is optimal. This scenario occurs when both types of drivers offer similar service quality, their costs are low, and the externalities are relatively large. In such case, the platform can provide high externalities to consumers at a low cost, with minimal variation in service levels between the two types of drivers.

If the retained cost of $H$ type drivers is high, strategy $M$ may not be the optimal for the platform. In such case, the platform will consider implementing a divisional service type approach.

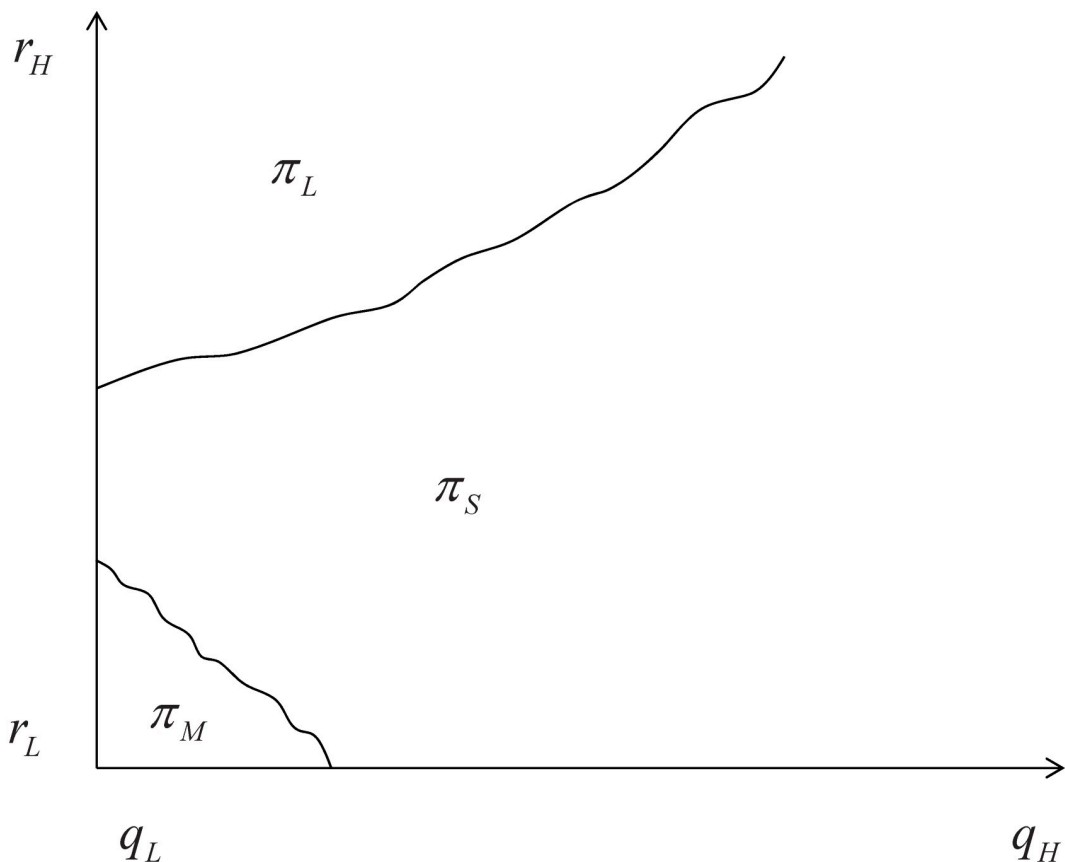

**Fig 3. Case with high externality and low retained cost of L-type drivers.**

This is because although both types of drivers have high external effects, the cost for retaining $H$ type drivers is relatively high. In order to improve profit margins, the platform can implement differential pricing.

When there is little difference in service levels between the two types of drivers, the platform will only attract $L$ type drivers. This is because the cost for differentiating service types and the potential profit margin is low in this situation. Therefore, the platform will choose to focus on attracting only $L$ type drivers.

However, if the retained cost for $H$ type drivers is high and both the external effect ($r_H$) and service quality ($q_H$) are high, the platform will opt for the differentiated service type approach once again.

That is, when the external effect is high and the retained cost of $L$ type drivers is low, the platform will focus on attracting and retaining $L$ type drivers. This is because the cost of retaining $H$ type drivers is much higher and the service quality between the two types is similar. Therefore, in this scenario, the platform will only use $L$ type drivers. However, in other cases where the retained cost of $H$ type drivers is not significantly higher than $L$ type drivers, and there is a difference in service quality, the platform will consider providing differentiated services. This is because the external effects are large, and the platform can cater to consumers' different service preferences. By offering different service types, such as $H$ type services at a higher price for consumers with higher service preferences, the platform can achieve optimal profits through differentiated pricing.

Next, we examine a scenario where the external effects increase and the unit retained cost of L-type drivers is relatively high. In this situation, when the external effects are significant and both $r_L$ and $r_H$ are large, it is necessary to compare the profitability of H-type drivers ($\pi_H$) and L-type drivers ($\pi_L$)

$$
\begin{aligned}
\pi_L - \pi_H \quad &= \frac{\alpha_L - \alpha_H}{2} + \frac{(\alpha_L)^2}{4q_L} - \frac{(\alpha_H)^2}{4q_H} - \frac{q_H - q_L}{4} \\
&+ r_H N_H - r_L N_L
\end{aligned}
$$

Based on the above analysis, it is important to discuss the difference in relative value between $\alpha_L$ and $\alpha_H$.

**Proposition 4.3** *let $\bar{\alpha}$ represent a certain value, $\bar{r}$ denote a specific threshold, and $\Delta\alpha$, $\Delta q$, $\Delta r$ represent incremental changes in external effect, quantities and costs. It is assumed that $\alpha_L$ is greater than or equal to $\bar{\alpha}$. If $r_L$ is greater than or equal to $\bar{r}$ and the external effects mitigation of H-type drivers is not significantly worse than L-type drivers ($\alpha_H - \alpha_L < \Delta\alpha$), the following conditions apply*:

*(1)When $r_H - r_L \leq \Delta r$, the optimal choice is $\pi_H$.*

*(2)When $r_H - r_L \geq \Delta r$ and $q_H - q_L \leq \Delta q$, the optimal choice is $\pi_L$.*

*(3) When $r_H - r_L > \Delta r$ and $q_H - q_L > \Delta q$, the optimal choice is $\pi_H$.*

Although the result aligns with proposition 2, the underlying factors driving this decision differ. When the external effects are minimal, the platform selects H type drivers primarily based on service quality considerations. By choosing H type drivers under such circumstances, the platform aims to enhance the overall service experience for consumers and attract them by providing better utilities. However, in scenarios where the network effect is sufficiently large but the differentiation between H type and L type drivers is minor, and the cost of retaining L type drivers is high, the platform's choice of H type drivers is predominantly influenced by service cost considerations. The platform is willing to invest more in attracting H type drivers as it improves service quality and ultimately leads to increased profitability through higher pricing.

The three propositions mentioned above illustrate a common understanding that platforms may evaluate both the cost and quality of service when making decisions. If the platform finds that the difference in service quality between H type and L type drivers is minimal, but the cost of retaining H type drivers is significantly higher, the platform may choose to only attract L type drivers to participate. On the other hand, if there is a substantial difference in service quality between the two types of drivers, the platform will consider involving H type drivers in providing services, either alone or in conjunction with L type drivers.

This can be referred to as the "cost-performance ratio" of H type drivers' participation. When the service quality provided by H type drivers is low and the retention cost is high, the "cost-performance ratio" is low. In such cases, the platform does not allow H type drivers to participate. However, if H type drivers can offer higher service quality at an acceptable retention cost, they have a high "cost-performance ratio". In these situations, the platform will allow them to participate.

Other cases are difficult to judge, and specific instances can be elucidated using numerical examples.

## 5 Numerical analysis

The theoretical conclusions obtained in the previous section offer optimal decisions given specific conditions of different network externalities. However, it is crucial to comprehend how the process of optimal decision-making changes with varying degrees of network externalities. To complement the theoretical findings, numerical examples can be analyzed to provide a more comprehensive understanding. By simulating different scenarios with different levels of network externalities, we can illustrate the optimal decisions based on specific parameters and assumptions used in the analysis.

In the model, we have several parameters: $r_i$, $q_i$, $N_i$, $N$, $\alpha_i$, and $\alpha_M$, where $i = L, H$. To supplement the conclusions, we will set specific values for these parameters and obtain the results of the example.

Based on the hypothesis of the model, externalities are positively correlated with the number of service providers. Many studies assume a linear relationship between externalities and the number of providers, which is based on Armstrong (2006) [7]. However, this paper introduces a different relationship between user waiting time and the number of providers, assuming a relationship of the form $\alpha_i = c(N_i)^{0.8}$, $\alpha_M = c^*N^{0.8}$. In this relationship, the variable c represents the unit network externality, which quantifies the change in external effects when the number of providers increases by one unit. This relationship demonstrates that the rate of change of external effects initially increases with the number of providers, but eventually decreases, as discussed in Bai (2018) [28].

While setting the numerical parameters, we will let $q_L$ and $r_L$ be fixed, and $r_L$ has high and low, two levels, other parameters including $N$, $N_L$, $N_H$, c are corresponding with high and low levels. Then, in the case of a set of parameters given above, the iteration of $q_H$ and $r_H$ will be carried out within a certain numerical range, and the optimal profits of the four cases are compared, so as to determine which case is optimal. While setting the numerical parameters, we will fix $q_L$ and $r_L$ at their respective values, while $r_L$ will have two levels: high and low. Other parameters, including $N$, $N_L$, $N_H$, and c, will correspond to high and low levels as well. Next, we will conduct iterations for $q_H$ and $r_H$ within a certain numerical range, given the above set of parameters. We will compare the optimal profits of the four cases to determine which case is optimal.

Specifically, we set the following values for the parameters: $N = 3$, $q_L = 1$. We will consider two scenarios for $r_L$, c, $N_L$ and $N_H$ respectively: one with a high value and one with a low value [36]. Based on the relationship between externalities and the number of service providers, $\alpha_i = c(N_i)^{0.8}$, we will iterate over different values of $q_H$ and $r_H$ and compare the optimal profits to determine which case is optimal. The detailed parameters can be seen in the Table 4.

When each group of the aforementioned parameters is fixed, we set the step size of $q_H$ (quantity provided by the high level) to 1 and the step size of $r_H$ (price at the high level) to 0.01. We iterate over a range of values for $q_H$ and $r_H$, with the maximum values being $Max(q_H) = 5$ and $Max(r_H) = 0.5$.

First, let's examine the scenario where $r_L$ is at a medium or low level. We will consider two cases that depend on the relative external effect with different levels of H and L. In the example where the external effect difference between the two types of services is considered, with $N_L = 1.6$, $N_H = 1.4$, we observe the changes in optimal decisions as c increases.

We can also observe similar patterns and changes in the optimal decision areas for other scenarios.

Figs 4 and 5 demonstrate that when $r_L$ is low, the change trends are similar regardless of whether the difference in external effects between the two types of services is high or low. In both figures, fig (a) shows that under the condition of relatively small external effects, when

**Table 4. Parameter settings.**

| | $N=3; q_L=1; max(q_H)=5; max(r_H)=0.5$ | | | | | | | |
|---|---|---|---|---|---|---|---|---|
| $r_L$ | 0.01 | 0.01 | 0.01 | 0.01 | 0.025 | 0.025 | 0.025 | 0.025 |
| $c$ | 0.1 | 0.1 | 0.4 | 0.4 | 0.1 | 0.1 | 0.4 | 0.4 |
| $N_L$ | 2.5 | 1.5 | 2.5 | 1.5 | 2.5 | 1.5 | 2.5 | 1.5 |
| $N_H$ | 0.5 | 1.4 | 0.5 | 1.4 | 0.5 | 1.4 | 0.5 | 1.4 |
| $\alpha_M$ | 0.241 | 0.241 | 0.953 | 0.953 | 0.241 | 0.241 | 0.953 | 0.953 |
| $\alpha_L$ | 0.208 | 0.145 | 0.833 | 0.583 | 0.208 | 0.145 | 0.833 | 0.583 |
| $\alpha_H$ | 0.057 | 0.131 | 0.23 | 0.524 | 0.057 | 0.131 | 0.23 | 0.524 |
| | $N=3; q_L=1; max(q_H)=5; max(r_H)=0.5$ | | | | | | | |
| $r_L$ | 0.05 | 0.05 | 0.05 | 0.05 | 0.1 | 0.1 | 0.1 | 0.1 |
| $c$ | 0.1 | 0.1 | 0.4 | 0.4 | 0.1 | 0.1 | 0.4 | 0.4 |
| $N_L$ | 2.5 | 1.5 | 2.5 | 1.5 | 2.5 | 1.5 | 2.5 | 1.5 |
| $N_H$ | 0.5 | 1.4 | 0.5 | 1.4 | 0.5 | 1.4 | 0.5 | 1.4 |
| $\alpha_M$ | 0.241 | 0.241 | 0.953 | 0.953 | 0.241 | 0.241 | 0.953 | 0.953 |
| $\alpha_L$ | 0.208 | 0.145 | 0.833 | 0.583 | 0.208 | 0.145 | 0.833 | 0.583 |
| $\alpha_H$ | 0.057 | 0.131 | 0.23 | 0.524 | 0.057 | 0.131 | 0.23 | 0.524 |

there is little difference between $r_H$ and $r_L$, the platform will prioritize drivers with high service quality. As the difference $r_H - r_L$ increases, the platform will prioritize drivers with high service quality. Additionally, if the difference between $q_H$ and $q_L$ is small, the platform will only utilize drivers with a low service quality level. On the other hand, if the difference between $q_H$ and $q_L$ is large, the platform will only utilize drivers with a high service quality level. These patterns highlight the importance of both the external effect and service quality in determining the optimal decisions of the platform. Mainly, the platform requires H-type drivers to enhance service quality and attract user participation.

Fig (b) illustrates that as $r_L$ is at a lower level, the optimal decisions change with the increase in $r_H$ and $q_H$ from small to large. As the external effect increases from (a) to (c), the platform's optimal decisions shift from H to S at a higher value of $q_H$. With a further increase in the external effect, the critical value of $q_H$ for the optimal decision to shift from H to S gradually moves to the left, eventually leading to the complete disappearance of the optimal decision H, as depicted in Fig (c).

In simpler terms, when the external effects of the L and H types of service become similar, an increase in external effects causes the platform's optimal decision to shift from H to S, starting with high service quality. This is because when the external effect difference is small, the platform needs a greater service level difference to justify implementing differential pricing. As the external effect value continues to increase, the optimal differentiation level $q_H$ gradually decreases. When the external effect becomes sufficiently large, the platform will consider the costs and decide whether to implement differentiated pricing or provide services only with L type drivers.

Next, when $r_L$ is at a medium or low level and the externalities of the two types of services are significantly different, it is reflected as $N_L = 2.5$ and $N_H = 0.5$ in Fig 6. As $c$ increases, we can observe changes in the optimal decision of the platform.

Similar case can be observed in Fig 7. The two examples demonstrate that when $r_L$ is at a medium or low level and there is a significant disparity in the external effects of H and L type services, the optimal decision of the platform shifts from H to S strategy as the external effects increase. This transition occurs starting from a point where the difference in quality between the two types is relatively low. In this case, the relatively high external effects can make a

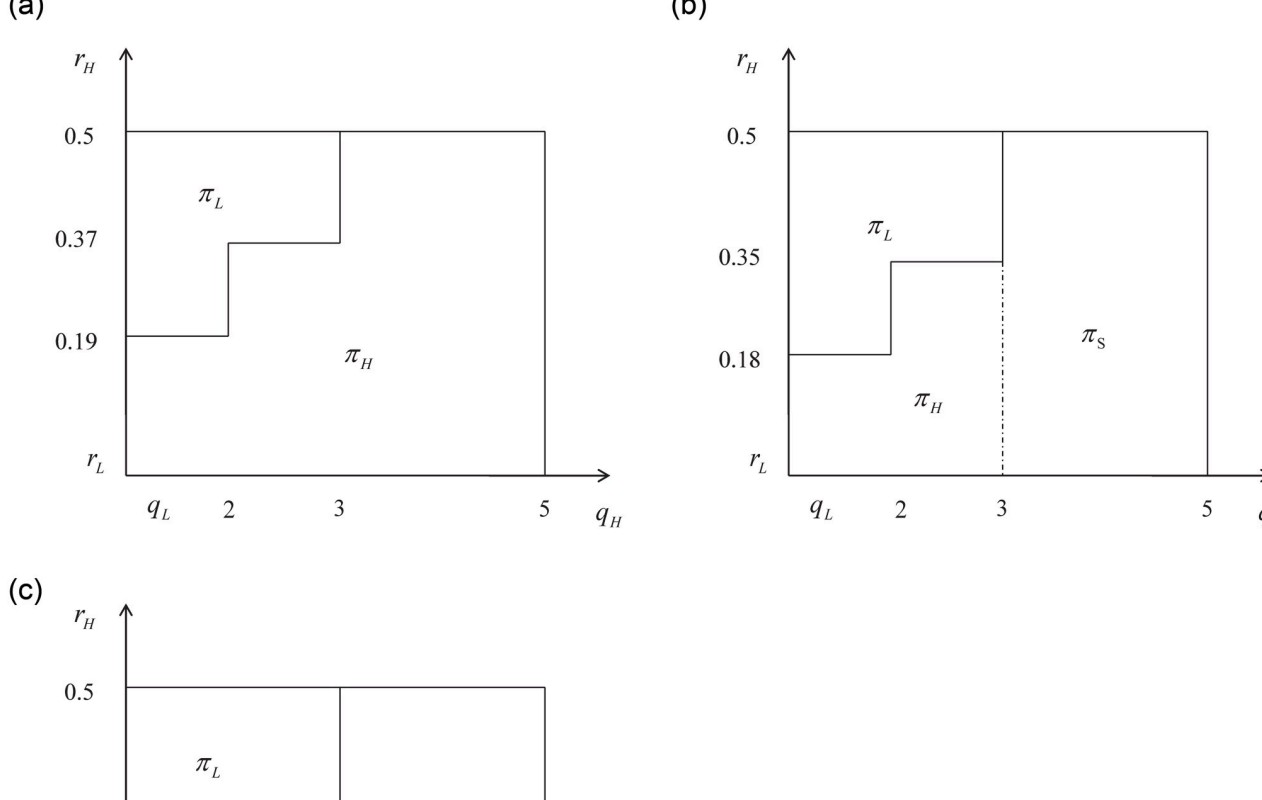

**Fig 4.** $r_L = 0.01; q_L = 1; N_L = 1.6; N_H = 1.4; max(r_H) = 0.5; max(q_H) = 5.$

significant impact on the decision-making process. The platform can implement differentiated pricing when the two types of service levels are relatively similar. While $q_L$ remains fixed, the optimal differentiated $q_H$ will gradually increase with the gradual increase in the external effect value.

From the previous discussions, it becomes evident that the platform takes into account both the cost and service quality when making decisions. When the value of $r_H$ is similar to that of $r_L$, the platform can effectively attract H type drivers to participate. However, if the quality level $q_H$ is close to $q_L$, the platform may not find it meaningful to implement differential pricing. In such cases, the platform may choose to mix drivers together and provide only a single service. In the scenario where the externalities are significant, the platform has the opportunity to charge a higher price. Additionally, if there is a substantial difference between the service quality levels $q_H$ and $q_L$, the platform can allow both types of drivers to provide services separately and implement differentiated pricing. As the value of $r_H$ increases, differentiated

(a)

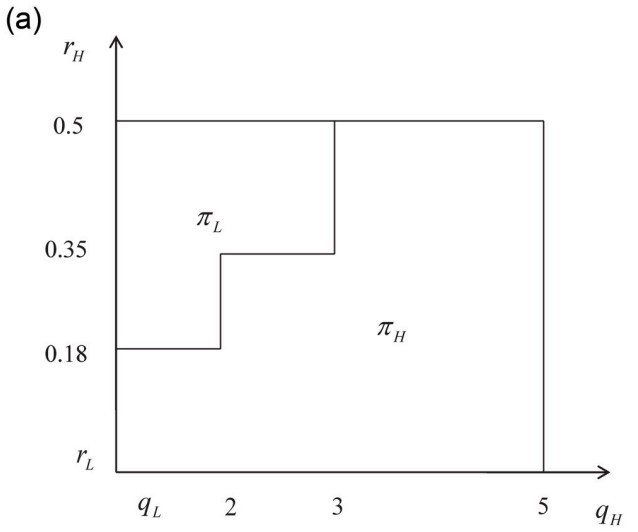

(b)

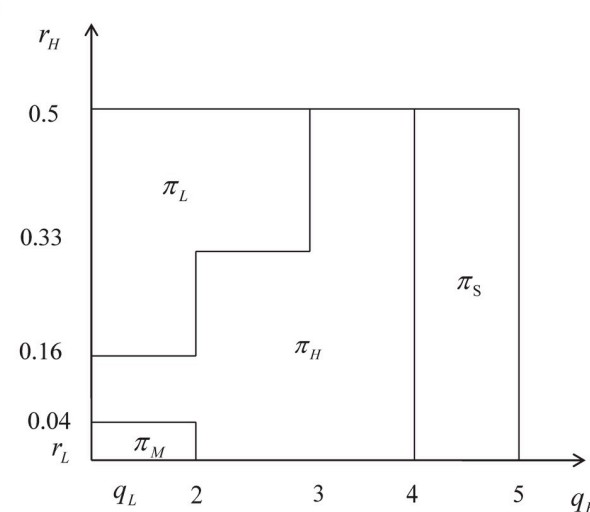

(c)

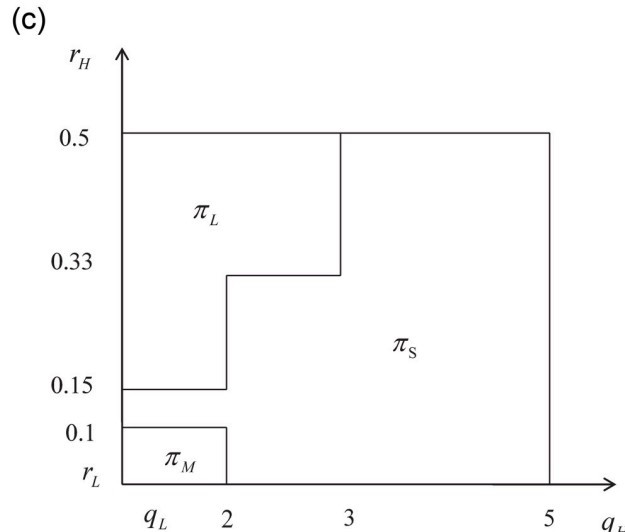

**Fig 5.** $r_L = 0.025$; $q_L = 1$; $N_L = 1.6$; $N_H = 1.4$; $max(r_H) = 0.5$; $max(q_H) = 5$.

pricing remains optimal within a certain range. If the value of $r_H$ becomes excessively high, the platform must carefully evaluate the "cost-performance ratio" of introducing H-type drivers to provide services. If the "cost-performance ratio" is low, indicating that the benefits of introducing H-type drivers are not significant enough, the platform may choose to have only L-type drivers provide services. However, if the "cost-performance ratio" is high, indicating that the benefits outweigh the costs, the platform will still aim to attract H-type drivers and provide differentiated services.

Fig (b) in the previous figures illustrate the transition from (a) to (c) as the external effect value increases. In this transition, the optimal decisions change from H/L to M/S/L, representing a shift in the platform's decision-making process. When $r_L$ is at a low level, the optimal decision of the platform changes as the external effect value increases. If there is a small difference between the external effects of H and L type services, the optimal decision starts from the

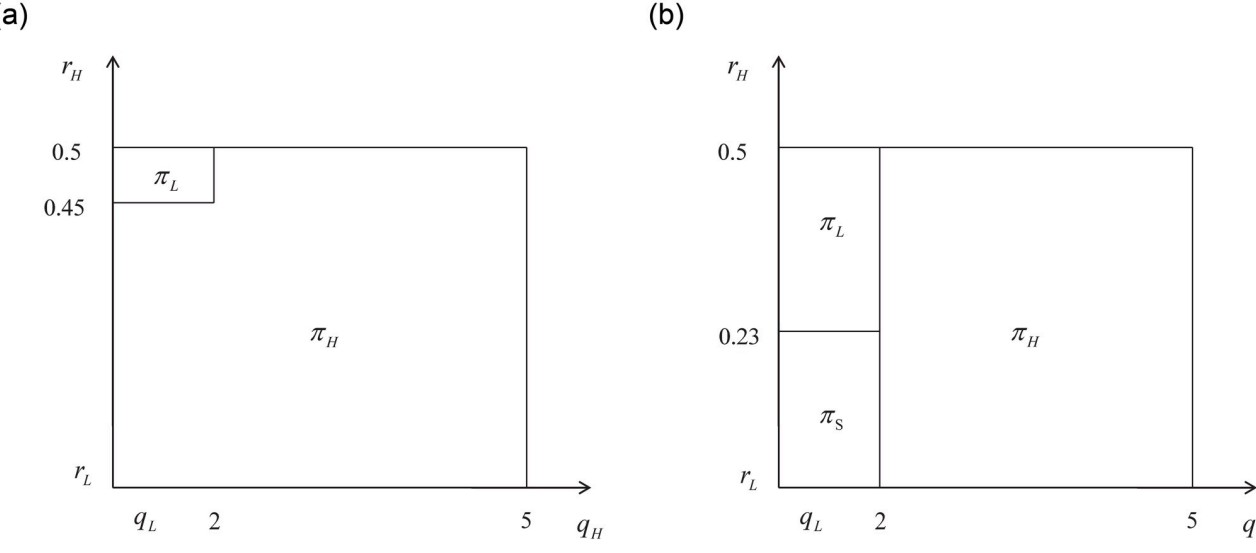

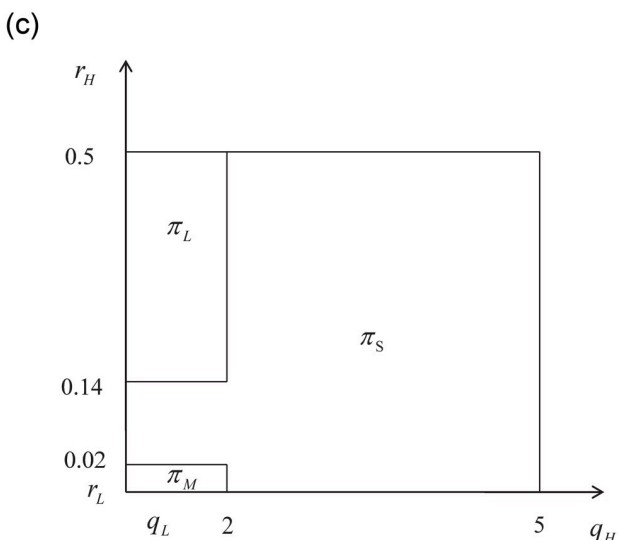

**Fig 6.** $r_L = 0.025$; $q_L = 1$; $N_L = 2.5$; $N_H = 0.5$; $max(r_H) = 0.5$; $max(q_H) = 5$.

point with high service quality of H type. However, if there is a large difference between the external effects of H and L type services, the optimal decision changes from H to S with the increase of external effects, starting from the point with low service quality of H type. This means that the platform can adjust its offerings based on the varying levels of service quality and external effects, catering to the different needs and preferences of its users.

When the value of $r_L$ is high, the optimal decision for the platform is to only select either H type or L type drivers, regardless of the magnitude of the external effects. This finding confirms and supplements Proposition 3.4. Specific numerical examples are as Figs 8 and 9.

Figs 8 and 9 illustrate the optimal decisions of the platform under the condition of high $r_L$, where the relative value difference of external effects is large or small. Comparing fig (a) in both sets of graphs, we can observe that when the absolute value of the external effect is small

(a)

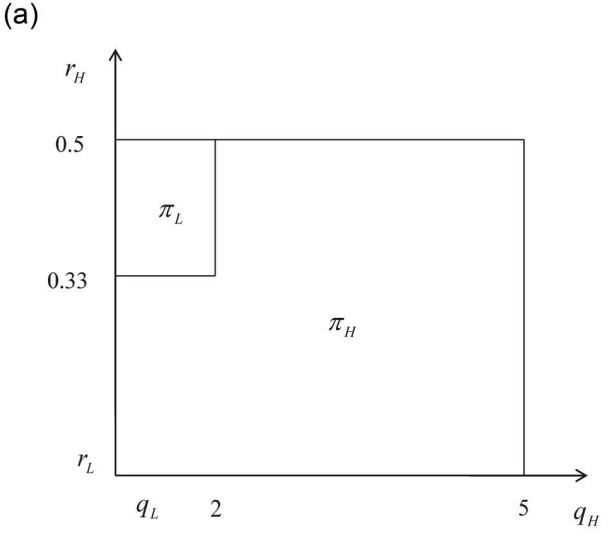

(b)

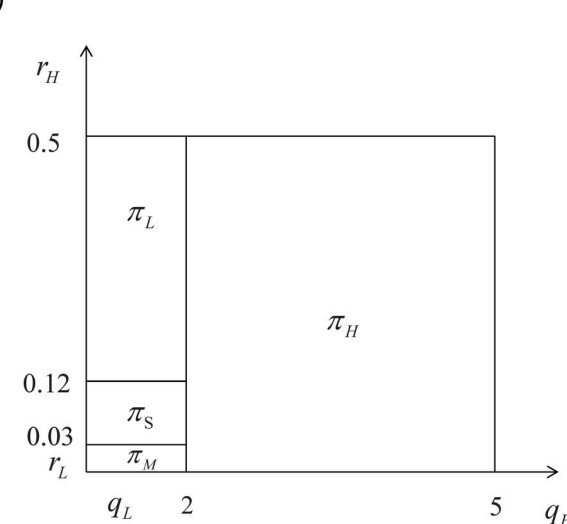

(c)

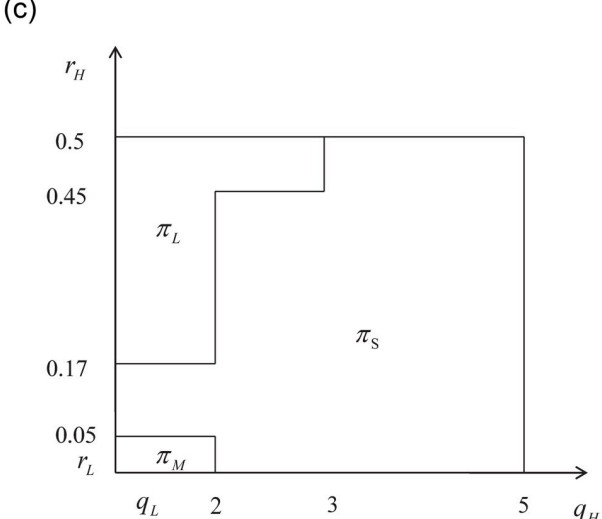

**Fig 7.** $r_L = 0.05$; $q_L = 1$; $N_L = 2.5$; $N_H = 0.5$; $max(r_H) = 0.5$; $max(q_H) = 5$.

and the relative value difference is large, the optimal decision of the platform remains the same: selecting only H type drivers to provide services. This is true even if the absolute value of the external effect in (a) of Fig 9 is smaller than that of Fig 8. This is because high external effects caused by L type drivers result in increased retention costs for the platform. Therefore, even if the platform only selects H type drivers to participate, it can still minimize the external effects and maximize its profits. Consequently, the platform can only serve a portion of the market.

Based on the analysis above, it is evident that when the network externality of low-quality services is low, the platform will only be able to attract high-quality drivers to participate if the service quality and cost are both high and within an acceptable range. Consequently, under the condition of low-quality service with low network externalities, the platform will have to forgo a portion of the market and only high-quality drivers will be able to cater to a certain segment

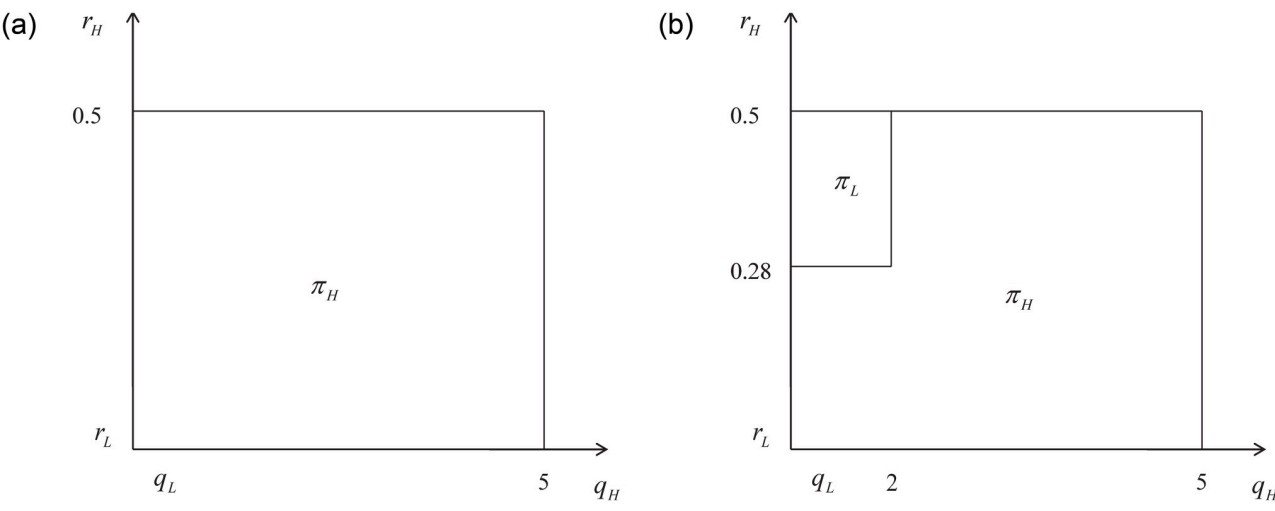

**Fig 8.** $r_L = 0.1$; $q_L = 1$; $N_L = 2.5$; $N_H = 0.5$; $max(r_H) = 0.5$; $max(q_H) = 5$.

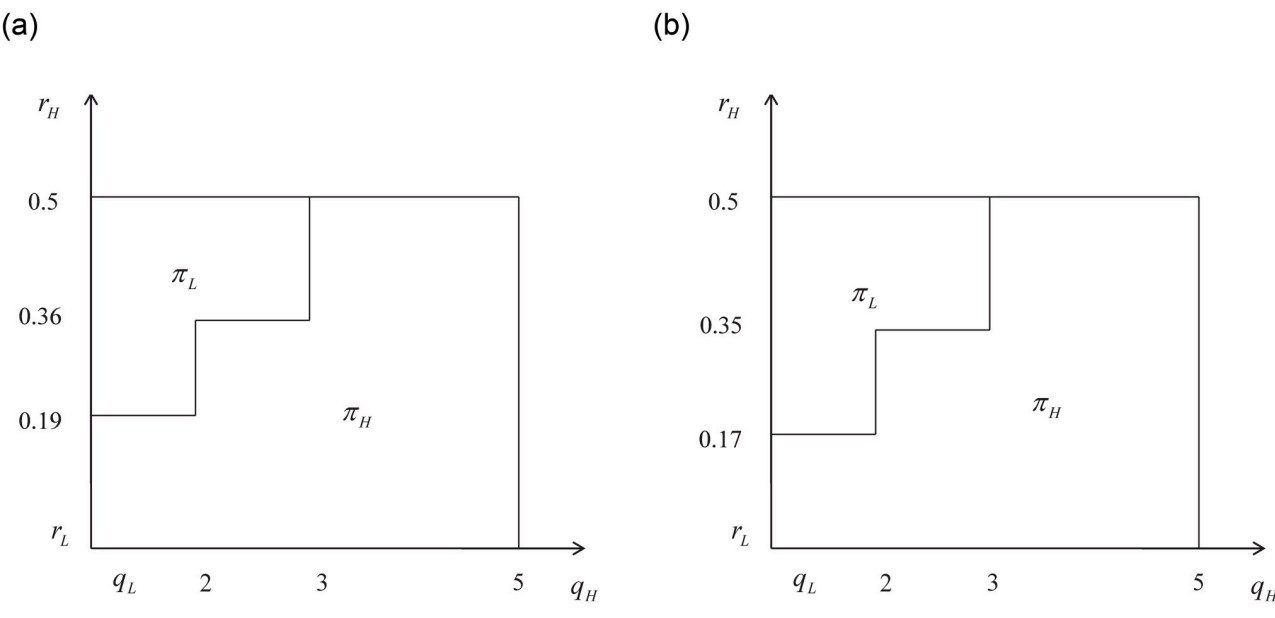

**Fig 9.** $r_L = 0.1$; $q_L = 1$; $N_L = 1.6$; $N_H = 1.4$; $max(r_H) = 0.5$; $max(q_H) = 5$.

of customers. This finding further supports the conclusion made by Zhong et al. (2020) that it is not always optimal for a platform to fulfill as many consumer orders as possible.

## 6 Managerial implication

The findings of this study have several managerial implications for platform companies.

Firstly, we find that platforms can measure the cost and quality of service to determine which type of driver to engage as service providers. If there is little difference in service quality between low and high-quality drivers, but a significant difference in cost, the platform can choose to attract only low-quality drivers to participate. On the other hand, if there is a

substantial gap in service quality between the two types of drivers, the platform may consider allowing high-quality drivers to participate and conduct a specific analysis to determine whether they should participate alone or alongside low-quality drivers. We can refer to this as the "cost-performance ratio" of high-quality drivers' participation. In situations where high-quality drivers incur high costs but deliver low service quality, the resulting "cost-performance ratio" is low. As a consequence, the platform can opt not to permit the participation of these drivers. In another scenario, despite the higher operational costs associated with high-quality drivers, their ability to consistently provide superior service quality results in a favorable "cost-performance ratio". Consequently, the platform can choose to allow them to participate.

Then, in numerical analysis, it becomes evident that the platform can consider cost, network externality, and service quality in determining the appropriate timing for implementing differentiated services. When the cost of retaining low-quality drivers is moderate or low, and if the externalities of both types of services are relatively similar, the platform should introduce differentiated pricing when there is a significant disparity between the two service levels. As the value of external effects gradually increases, the optimal differentiation can be achieved with lower service quality. In cases where there is a significant difference in the external effects of two types of services, the platform can choose to implement differentiated pricing even when the service levels are similar. As the value of external effects increases gradually, the service quality needed to achieve optimal differentiation also increases gradually. However, when the cost of retaining low-quality drivers is high, the platform's optimal decision is to involve either only high-quality or only low-quality drivers, depending on their respective costs. If the cost of high-quality drivers is only slightly higher than that of low-quality drivers, the platform can attract drivers with a high service level to participate. However, if the cost of high-quality drivers is significantly higher, the platform can only attract low-quality drivers to participate.

Overall, the findings of this study emphasize the importance of understanding the dynamics of network externality and service quality in the platform economy. By recognizing the impact of external effects and differentiating service quality, platforms can improve their competitiveness. By attracting the right type of drivers, platforms can enhance the overall customer experience. This study highlights the significance of these factors and encourages platforms to implement effective strategies to stay ahead in the market.

## 7 Conclusion

In this paper, the inclusion of both cross-network externalities and service quality preference in the user utility function highlights the importance of considering these factors in platform decisions. This emphasizes the need for platforms to carefully consider the impact of network effects and service quality on user behavior to maximize profitability and provide a superior customer experience. We assume the existence of two types of drivers: high and low type. This allows the platform to explore different strategies, such as attracting one type or both types of drivers, pricing them differently or together with only one type of service. By comparing the profits under each decision in different conditions, the paper provides insights into the optimal strategy for the platform.

First of all, the platform's decision on driver quality depends on the trade-off between cost and service quality, as well as consumer preferences and their willingness to pay for higher quality. When the externality of drivers is relatively small, the platform should consider the "cost-performance ratio" in relation to the retention cost and service quality of the two types of drivers, regardless of whether the unit retention cost of low-quality drivers is high or low. In other words, if the unit cost difference between the two types of drivers is not significant, the platform can afford a higher cost and opt for high-quality drivers. However, as the cost for

such drivers increases, the platform will continue to attract only low-quality drivers if the "cost performance" of high service quality is not sufficiently high.

In addition, In the case of a large external effect and high retention cost of low-quality drivers, the overall result may be similar to the previous case, but the internal driving factors are different. In the previous case, where the external effect was small, improving service quality was a measure to attract consumers to participate in the service. However, in this case, where the external effect is larger and there is a high retention cost associated with low-quality drivers, the cost factor becomes the main consideration for the platform.

Moreover, when the external effect is significant and the retention cost of low-quality drivers is low, the platform may consider adopting a strategy where all drivers serve with price discrimination. In this case, if service quality of high-quality drivers is relatively high, the platform can implement price discrimination. This approach ensures that consumers who value and are willing to pay for high-quality service can access it, while those who are more price-sensitive can still find lower-cost options. However, if the cost of high-quality drivers is too high and there is minimal difference in service quality between high and low-quality drivers, the platform may choose to abandon high-quality drivers and solely attract low-quality drivers to participate.

Finally, some conclusions have been derived from the numerical examples, which complement the theoretical findings from a dynamic perspective. The retention cost of low-quality drivers is divided into two categories: medium/low level and high level.

When the cost of retaining low-quality drivers is at a medium/low level and the external impact of both service types is relatively similar, the platform should adopt differentiated pricing if there is a significant difference in service levels. However, as the external effect values increase, the optimal differentiation of service quality gradually decreases. When there is a significant discrepancy in the external effects of the two types of services, the platform can introduce differentiated pricing even if the service levels are relatively similar. As external effect values increase, the optimal differentiation of service quality also increases.

In addition, when the retention cost of low-quality drivers is higher, regardless of the absolute value of the external effect, the platform's optimal decision is to attract either high-quality or low-quality drivers. When the cost of high-quality drivers is not significantly higher than that of low-quality drivers, the platform may be more inclined to attract drivers with a higher service level. Otherwise, the platform may only be able to attract drivers with low quality to participate.

## Author Contributions

**Conceptualization:** Lina Ma.

**Data curation:** Qiang Wei.

**Formal analysis:** Lina Ma.

**Investigation:** Qiang Wei.

**Methodology:** Lina Ma.

**Project administration:** Zhijie Tao.

**Supervision:** Zhijie Tao.

**Validation:** Lina Ma.

**Visualization:** Lina Ma.

**Writing – original draft:** Lina Ma.

**Writing – review & editing:** Lina Ma.

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
