## [Decision Letter · Decision Letter 0]

6 Jun 2023

PONE-D-23-03432On-demand ride service platform with differentiated servicesPLOS ONE

Dear Dr. Wei,

Thank you for submitting your manuscript to PLOS ONE. After careful consideration, we feel that it has merit but does not fully meet PLOS ONE’s publication criteria as it currently stands. Therefore, we invite you to submit a revised version of the manuscript that addresses the points raised during the review process.

We look forward to receiving your revised manuscript.

Kind regards,

Juan Carlos Rocha Gordo

Academic Editor

PLOS ONE

Journal Requirements:

2. Thank you for including your ethics statement:  "N/A".  

a. For studies reporting research involving human participants, PLOS ONE requires authors to confirm that this specific study was reviewed and approved by an institutional review board (ethics committee) before the study began. Please provide the specific name of the ethics committee/IRB that approved your study, or explain why you did not seek approval in this case.

b. Please provide additional details regarding participant consent. In the ethics statement in the Methods and online submission information, please ensure that you have specified (1) whether consent was informed and (2) what type you obtained (for instance, written or verbal, and if verbal, how it was documented and witnessed). If your study included minors, state whether you obtained consent from parents or guardians. If the need for consent was waived by the ethics committee, please include this information.

Additional Editor Comments (if provided):

Dear authors,

Please find attached the reviewer recommendations of one reviewers. Unfortunately more than 25 people declined to review your manuscript and I had to make a decision with only one review. To give a fair treatment I also read and reviewed your work.

In its current form the manuscript cannot be published, although the study is interesting and some valuable results are presented. The paper suffers from a poor use of the English language, both in terms of grammar and spelling. The paper is also disorganized and repetitive. The purpose of the manuscript needs to come up front (by the introduction) and it would be ideal to have an example of the an application problem of your question, given that the readership of PlosONE is an interdisciplinary audience. The figures are not of enough quality for publication, I could barely read them on the pdf. The submission form indicates data are available but I could not find the code to reproduce the figures. I do not understand why you get wiggling in some of your figures.

I strongly recommend you restructure your paper to present the problem and modeling work on a succinct fashion. The literature review section should summarize the key contributions of previous work, not list every single paper you've read nor name them all. For example, you can group papers by their type of contribution, state what the contribution is, and make a reference to multiple papers (x,y,z); instead of writing separate sentences by paper starting the sentence with the authors (x said A and B. Y found B. Z also found B). For future submissions to any journal, the cover letter should not be a copied-pasted version of the abstract.

Please find attached the recommendations of Reviewer 1. I believe the reviewer's suggestions will help you improve your work and prepare it for resubmission.

Best regards,

Juan Rocha

Reviewers' comments:

Reviewer's Responses to Questions

**Comments to the Author**

1. Is the manuscript technically sound, and do the data support the conclusions?

Reviewer #1: Yes

2. Has the statistical analysis been performed appropriately and rigorously? 

Reviewer #1: N/A

3. Have the authors made all data underlying the findings in their manuscript fully available?

Reviewer #1: Yes

4. Is the manuscript presented in an intelligible fashion and written in standard English?

Reviewer #1: Yes

5. Review Comments to the Author

Reviewer #1: This paper explores the pricing strategy considering the influence of consumer preference heterogeneity and different service types of drivers with network externalities. The research content is relatively complete and the writing is standardized; however, the paper still needs minor revision before it is accepted and published. My detailed comments are as follows:

1. Some of the expressions in the text are too repetitive, and it is suggested to make appropriate deletions to make the writing clearer and increase readability.

2. The assumptions of the model are too idealistic, and the driver's income equation considers limited factors. The income is also affected by factors such as demand time, and distance etc., which lack corresponding literature support.

3. The parameter settings in the numerical analysis section lack necessary practical support and corresponding data sources, which reduces the realism of the paper.

4. The first and third conclusions, as well as the second and fourth conclusions, are quite similar and could be merged. It is recommended to delete overly intuitive conclusions and add more managerial insights.

5. Additionally, I recommend that the authors consider delegating the writing to a professional linguistic agency to improve the language expression.

6. PLOS authors have the option to publish the peer review history of their article (what does this mean?). If published, this will include your full peer review and any attached files.

Reviewer #1: No

---

## [Author Response · Author response to Decision Letter 0]

5 Sep 2023

Response to Reviewer’s and Editor’s Comments:

We rewrite the whole paper, including abstract, the introduction, the literature and the conclusion. The repetitive expressions are either deleted or reorganized. We also add some corresponding literature support to the assumptions of model. In addition, we also update the references.

1. Some of the expressions in the text are too repetitive, and it is suggested to make appropriate deletions to make the writing clearer and increase readability.

RESPONSE: Thank you for pointing out the issue of repetitiveness in our text. We appreciate your suggestion to make appropriate deletions in order to improve clarity and readability.

Example 1:

Abstract

The rapid growth of on-demand ride service platforms has made it increasingly important for these platforms to efficiently match services by understanding driver characteristics and consumer preferences. This paper aims to investigate the pricing strategy by considering the impact of consumer preference heterogeneity and the different service types offered by drivers. The findings of this study reveal the need for the platform to strike a balance between service cost and the benefits of high-quality drivers, which can be referred to as the “cost-performance ratio”.If the “cost-performance ratio” that attracts high-quality drivers is high, the platform will attract high-quality drivers or drivers of all types to participate while offering differentiated services. Otherwise, the platform will only provide services through low-quality drivers. Furthermore, the platform will also consider when to offer differentiated services based on network externalities and service quality. When the network externalities of the two types of services are similar, the platform will differentiate them based on service quality differences. Overall, considering consumer preference heterogeneity, drivers of service types, and network externalities, this paper provides guidance for platforms to make optimal decisions that enhance their service offerings and improve overall customer satisfaction.

Key words: On-demand ride service platform; Driver service quality; Consumer service preference, Network externalities

Example 2:

Conclusion

In this paper, the inclusion of both cross-network externalities and service quality preference in the user utility function highlights the importance of considering these factors in platform decisions. This emphasizes the need for platforms to carefully consider the impact of network effects and service quality on user behavior to maximize profitability and provide a superior customer experience. We assume the existence of two types of drivers: high and low type. This allows the platform to explore different strategies, such as attracting one type or both types of drivers, pricing them differently or together with only one type of service. By comparing the profits under each decision in different conditions, the paper provides insights into the optimal strategy for the platform.

First of all, the platform’s decision on driver quality depends on the trade-off between cost and service quality, as well as consumer preferences and their willingness to pay for higher quality. When the externality of drivers is relatively small, the platform should consider the“cost-performance ratio”in relation to the retention cost and service quality of the two types of drivers, regardless of whether the unit retention cost of low-quality drivers is high or low. In other words, if the unit cost difference between the two types of drivers is not significant, the platform can afford a higher cost and opt for high-quality drivers. However, as the cost for such drivers increases, the platform will continue to attract only low-quality drivers if the“cost performance”of high service quality is not sufficiently high.

In the case of a large external effect and high retention cost of low-quality drivers, the overall result may be similar to the previous case, but the internal driving factors are different. In the previous case, where the external effect was small, improving service quality was a measure to attract consumers to participate in the service. However, in this case, where the external effect is larger and there is a high retention cost associated with low-quality drivers, the cost factor becomes the main consideration for the platform.

Moreover, when the external effect is significant and the retention cost of low-quality drivers is low, the platform may consider adopting a strategy where all drivers serve with price discrimination. In this case, if service quality of high-quality drivers is relatively high, the platform can implement price discrimination. This approach ensures that consumers who value and are willing to pay for high-quality service can access it, while those who are more price-sensitive can still find lower-cost options. However, if the cost of high-quality drivers is too high and there is minimal difference in service quality between high and low-quality drivers, the platform may choose to abandon high-quality drivers and only attract low-quality drivers to participate.

Finally, some conclusions have been derived from the numerical examples, which complement the theoretical findings from a dynamic perspective. The retention cost of low-quality drivers is divided into two categories: medium/low level and high level. 

When the cost of retaining low-quality drivers is at a medium/low level and the external impact of both service types is relatively similar, the platform should adopt differentiated pricing if there is a significant difference in service levels. However, as the external effect values increase, the optimal differentiation of service quality gradually decreases. When there is a significant discrepancy in the external effects of the two types of services, the platform can introduce differentiated pricing even if the service levels are relatively similar. As external effect values increase, the optimal differentiation of service quality also increases.

In addition, when the retention cost of low-quality drivers is higher, regardless of the absolute value of the external effect, the platform’s optimal decision is to attract either high-quality or low-quality drivers. When the cost of high-quality drivers is not significantly higher than that of low-quality drivers, the platform may be more inclined to attract drivers with a higher service level. Otherwise, the platform may only be able to attract drivers with low quality to participate.

2.The assumptions of the model are too idealistic, and the driver's income equation considers limited factors. The income is also affected by factors such as demand time, and distance etc., which lack corresponding literature support.

RESPONSE: 

Thank you for your feedback regarding the assumptions and factors considered in our model. We appreciate your perspective on the idealistic nature of the assumptions and the limited factors included in the driver’s income equation.

You are correct in pointing out that there are additional factors, such as demand time and distance, that can affect a driver’s income. We understand the importance of considering the factors. At the beginning, we take different factors into consideration in the model, but we found it is difficult to calculate. In our research, we focused on natures of the factors that were supported by existing literature and were deemed relevant for our analysis. 

In light of your feedback, we revised our model description to explain these additional factors and ensure that we provide appropriate literature support for their inclusion. This will help strengthen the validity and applicability of our findings.

In many studies, the utility function for both users in a two-sided market is often assumed to be the same. However, considering the unique characteristics of users in on-demand ride service platform, this paper takes into account different factors for consumers and drivers participation. Consumers take the external effects, prices, and service quality of drivers into consideration, whereas drivers prioritize their income and the efficiency of receiving orders. To model the utility function for drivers, this paper draws on the form used by Zhong et al. (2020). It considers that drivers make their participation decision based on their income, which is influenced by the platform salary, the efficiency of receiving orders, and their own retention costs. This more comprehensive utility function captures the various factors that influence driver participation.

The platform classifying drivers into high quality (H-type) or low quality (L-type) based on certain characteristics is a common practice in many on-demand service platforms. This classification helps the platform differentiate between drivers and provide different levels of service to customers based on their preferences and willingness to pay. The unit salary given by the platform is and respectively. Driver’s revenue function is , is the number of consumers choosing i type, is the number of i type drivers, assume that every consumer create only one service demand, so, is the order volume of each driver. If 0, the driver will participate. 

3. The parameter settings in the numerical analysis section lack necessary practical support and corresponding data sources, which reduces the realism of the paper.

RESPONSE:

Thank you for your comment regarding the parameter settings in the numerical analysis section of our paper. We appreciate your concern about the lack of practical support and corresponding data sources for these settings, which can potentially reduce the realism of our study.

We understand the importance of ensuring our parameter settings align with real-world conditions. We apologize for not providing sufficient practical support and data sources in this section. To rectify this, we will conduct further research to gather accurate and relevant data that can better inform our parameter settings. This will allow us to improve the realism of our study and enhance the credibility of our results.

We appreciate your feedback and assure you that we will make the necessary revisions to address this concern. Your input is invaluable in helping us improve the quality and validity of our research.

In our paper the purpose of numerical analysis is mainly to verify and supplement the theoretical research part through the iteration of the data. The corresponding approach can be found in previous studies and we explain this in the paper.

The theoretical conclusions obtained in the previous section offer optimal decisions given specific conditions of different network externalities. However, it is crucial to comprehend how the process of optimal decision-making changes with varying degrees of network externalities. To complement the theoretical findings, numerical examples can be analyzed to provide a more comprehensive understanding. By simulating different scenarios with different levels of network externalities, we can illustrate the optimal decisions based on specific parameters and assumptions used in the analysis.

In the model, we have several parameters: , , , N, and , where i = L, H. To supplement the conclusions, we will set specific values for these parameters and obtain the results of the example.

Based on the hypothesis of the model, externalities are positively correlated with the number of service providers. Many studies assume a linear relationship between externalities and the number of providers, which is based on Armstrong (2006). However, this paper introduces a different relationship between user waiting time and the number of providers, assuming a relationship of the form ,. In this relationship, the variable c represents the unit network externality, which quantifies the change in external effects when the number of providers increases by one unit. This relationship demonstrates that the rate of change of external effects initially increases with the number of providers, but eventually decreases, as discussed in Bai (2018).

4. The first and third conclusions, as well as the second and fourth conclusions, are quite similar and could be merged. It is recommended to delete overly intuitive conclusions and add more managerial insights.

RESPONSE: 

Thank you for your suggestion regarding the merging of similar conclusions and the inclusion of more managerial insights in our paper. Upon reviewing the paper, we agree that some of the conclusions could be merged to avoid repetition and improve clarity. We carefully reevaluated our findings and streamline the conclusions accordingly.

Although the result(In the paper we omit the repetitive description of the result) aligns with proposition 2, the underlying factors driving this decision differ. When the external effects are minimal, the platform selects high type drivers primarily based on service quality considerations. By choosing high type drivers under such circumstances, the platform aims to enhance the overall service experience for consumers and attract them by providing better utilities. However, in scenarios where the network effect is sufficiently large but the differentiation between high type and low type drivers is minor, and the cost of retaining low type drivers is high, the platform’s choice of high type drivers is predominantly influenced by service cost considerations. The platform is willing to invest more in attracting high type drivers as it improves service quality and ultimately leads to increased profitability through higher pricing.

The three propositions mentioned above illustrate a common understanding that platforms may evaluate both the cost and quality of service when making decisions. If the platform finds that the difference in service quality between high type and low type drivers is minimal, but the cost of retaining high type drivers is significantly higher, the platform may choose to only attract low type drivers to participate. On the other hand, if there is a substantial difference in service quality between the two types of drivers, the platform will consider involving high type drivers in providing services, either alone or in conjunction with low type drivers.

This can be referred to as the “cost-performance ratio” of high type drivers’ participation. When the service quality provided by high type drivers is low and the retention cost is high, the “cost-performance ratio” is low. In such cases, the platform does not allow high type drivers to participate. However, if high type drivers can offer higher service quality at an acceptable retention cost, they have a high “cost-performance ratio”. In these situations, the platform will allow them to participate.

In addition, we understand the importance of providing clear and actionable insights for managers based on our research findings. So we added the managerial implications section to provide more detailed and specific recommendations for practitioners. 

Managerial implication

The findings of this study have several managerial implications for platform companies. 

Firstly, we find that platforms can measure the cost and quality of service to determine which type of driver to engage as service providers. If there is little difference in service quality between low and high-quality drivers, but a significant difference in cost, the platform can choose to attract only low-quality drivers to participate. On the other hand, if there is a substantial gap in service quality between the two types of drivers, the platform may consider allowing high-quality drivers to participate and conduct a specific analysis to determine whether they should participate alone or alongside low-quality drivers. We can refer to this as the “cost-performance ratio” of high-quality drivers’ participation. In situations where high-quality drivers incur high costs but deliver low service quality, the resulting “cost-performance ratio” is low. As a consequence, the platform can opt not to permit the participation of these drivers. In another scenario, despite the higher operational costs associated with high-quality drivers, their ability to consistently provide superior service quality results in a favorable “cost-performance ratio.” Consequently, the platform can choose to allow them to participate. 

Then, in numerical analysis, it becomes evident that the platform can consider cost, network externality, and service quality in determining the appropriate timing for implementing differentiated services. When the cost of retaining low-quality drivers is moderate or low, and if the externalities of both types of services are relatively similar, the platform should introduce differentiated pricing when there is a significant disparity between the two service levels. As the value of external effects gradually increases, the optimal differentiation can be achieved with lower service quality. In cases where there is a significant difference in the external effects of two types of services, the platform can choose to implement differentiated pricing even when the service levels are similar. As the value of external effects increases gradually, the service quality needed to achieve optimal differentiation also increases gradually. However, when the cost of retaining low-quality drivers is high, the platform’s optimal decision is to involve either only high-quality or only low-quality drivers, depending on their respective costs. If the cost of high-quality drivers is only slightly higher than that of low-quality drivers, the platform can attract drivers with a high service level to participate. However, if the cost of high-quality drivers is significantly higher, the platform can only attract low-quality drivers to participate.

Overall, the findings of this study emphasize the importance of understanding the dynamics of network externality and service quality in the platform economy. By recognizing the impact of external effects and differentiating service quality, platforms can improve their competitiveness. By attracting the right type of drivers, platforms can enhance the overall customer experience. This study highlights the significance of these factors and encourages platforms to implement effective strategies to stay ahead in the market. 

5.Additionally, I recommend that the authors consider delegating the writing to a professional linguistic agency to improve the language expression.

RESPONSE: 

Thank you for your suggestion to consider delegating the writing to a professional linguistic agency to improve the language expression in our paper. We understand the importance of clear and effective language in conveying our research findings.We have taken extra care in revising and editing our manuscript to improve the language expression and address the issues with grammar, sentence structure, or word choice. We also seek feedback from language experts and native English speakers to ensure the accuracy and fluency of our language. Thank you again for your recommendation. 

6.The purpose of the manuscript needs to come up front (by the introduction) and it would be ideal to have an example of the an application problem of your question, given that the readership of PlosONE is an interdisciplinary audience.

RESPONSE: 

Thank you for your valuable feedback. We agree that it is important to clearly communicate the purpose of our manuscript in the introduction. We will ensure that the introduction provides a concise and explicit statement of the research question and the objectives of our study.

Additionally, we understand the importance of providing an example application problem to engage and appeal to the interdisciplinary readership of PLOS ONE. We have incorporate relevant and illustrative examples in the introduction to demonstrate the practical implications and potential applications of our research.

Introduction

Platforms have revolutionized traditional business models by offering consumers convenient and efficient means to access a diverse array of products and services. For example, online shopping platforms like Amazon and Alibaba have changed the way people buy goods by offering a vast selection, competitive prices, and fast delivery. Third-party payment platforms like PayPal and Alipay have simplified online transactions and made it easier for people to make payments and transfer money securely. Sharing platforms like Uber and Airbnb have enabled individuals to monetize their assets, such as cars and spare rooms, by connecting them with people who need those services. These platform businesses have created new opportunities for entrepreneurs and individuals to generate income and have empowered consumers with more choices and convenience(Qin et al.,2016;Tian etal.,2018;Zhao et al.209;Pei et al.2021). The platform sharing economy is still evolving, and its impact on various industries and society as a whole is still being studied. It is crucial for policymakers, businesses, and individuals to understand and adapt to this new economic model to ensure its benefits are maximized while addressing any potential challenges that may arise.

The platform also handles the payment process, ensuring a secure transaction for both parties involved(Taylor,2018;Yu et al.2019). The platform’s operation is crucial for the success of the on-demand economy. It facilitates the connection between consumers and service providers, sets prices, ensures trust and safety, and provides the necessary technology for seamless transactions(Cui and Hu,2018). It typically includes mobile apps or websites that allow consumers to easily request rides and drivers to accept or decline requests. The platform is responsible for matching drivers with consumers based on various factors, such as location and availability(Chen,2020).

As a type of two-sided market, an on-demand ride service platform acts as an intermediary between drivers and consumers. Its distinguishing feature is the cross-network externality, where the utility of one user is influenced by the size of the other group(Armstrong,2006). Therefore, the initial goal of an on-demand service platform is to ensure a sufficient number of participants, as the cross-network externality is a distinguishing feature of two-sided markets. On one hand, when there is a high number of participating drivers, consumers are more likely to join the platform due to shorter waiting times. On the other hand, a high number of consumers increases driver participation in the platform due to the availability of numerous orders. Consequently, a mutually agreeable wage or price is established for trading purposes. The platform, acting as an intermediary, receives a percentage of the profits from each order, in accordance with the agreed arrangement for all parties involved.

In previous studies on platform differentiation, some scholars have linked differentiation with network effects, suggesting that in industries with network effects, network size is a more important factor of competition than quality(Baake and Boomy,2001). Later literature has defined service quality as network externalities(Zhao,et al.,2020). With the development of two-sided markets, both sides have heterogeneous preferences and features. Consumers’ preferences influence their participation in decision-making processes(Acheampong,2020). Firstly, consumers have varying preferences for service quality. Some are willing to pay a higher price for high-quality service, while others are willing to accept lower service quality at a lower price as long as the waiting time is appropriate. Additionally, drivers themselves have different levels of service quality, including the type of vehicles and diverse service levels provided by drivers(Ning et al.,2019).

With the booming of on-demand service platforms, it is indeed necessary for platforms to provide different experiences in order to carve out their own niche. Additionally, platforms need to cater to consumers with different service preferences by offering various types of services. In reality, platforms like Didi provide differentiated service types such as express, private car, and hitch. Each service type corresponds to a different type of driver, and consumers who choose these services generally have similar preferences. By differentiating services according to driver types, the platform can maximize the utility and benefits for all participants to a greater extent. For example, the platform can match high-quality services with high-quality preference consumers, thereby implementing price discrimination. So it becomes crucial for the platform to identify the different features or requirements of both sides (consumers and drivers) and find ways to achieve more accurate matching.

Based on consumers’ preferences and the varying levels of service provided by drivers, the platform aims to achieve more precise matching between consumers and providers(Chu and Manchanda,2017). One approach is to allow all drivers to offer a hybrid service, which can generate a pooling effect that is greater compared to when they provide separate services. The platform can also make different service decisions based on various conditions. For instance, if the number of consumers is not very large, the platform may consider having only a portion of the service providers offer their services to save costs. On the other hand, if the number of consumers is large, the service will be provided by all drivers, allowing the platform to benefit from high network externalities. To determine whether service differentiation is beneficial for platforms and when to implement it, as well as to identify the optimal type of driver to provide service under different conditions, this paper takes into account the characteristics of network externalities in the context of on-demand service platforms. It also considers the heterogeneity of consumers’ service preferences on the demand side and the different service types of drivers on the supply side, aiming to explore the optimal pricing decision for the platform.

7.The figures are not of enough quality for publication, I could barely read them on the pdf. The submission form indicates data are available but I could not find the code to reproduce the figures. I do not understand why you get wiggling in some of your figures.

RESPONSE: 

We apologize for the poor quality of the figures in the manuscript. Regarding the availability of data and code, we apologize for the oversight. We will ensure that all necessary data and code are made available upon publication. As for the wiggling observed in some of the figures, we will carefully investigate the cause of this issue. It is possible that it may be due to technical or data processing errors. We will thoroughly review the data analysis procedures and methodologies to identify and rectify any potential errors that may have led to this wiggling effect.

Thank you for bringing these concerns to our attention. We appreciate your feedback and will take the necessary steps to address these issues and improve the quality of our manuscript.

8.I strongly recommend you restructure your paper to present the problem and modeling work on a succinct fashion. The literature review section should summarize the key contributions of previous work, not list every single paper you've read nor name them all. For example, you can group papers by their type of contribution, state what the contribution is, and make a reference to multiple papers (x,y,z); instead of writing separate sentences by paper starting the sentence with the authors (x said A and B. Y found B. Z also found B).

RESPONSE: 

Thank you for your valuable suggestion. We appreciate your feedback on the structure and content of our paper. We have restructured the paper to provide a more succinct and cohesive overview of the problem and our modeling approach.

Regarding the literature review section, we have revised it to highlight the key contributions of previous work rather than providing an exhaustive list of every paper we have read. We grouped papers based on their type of contribution, summarize the main findings, and provide a reference to multiple papers to support our statements. This approach will help streamline the literature review section and make it more concise and reader-friendly.

2 Literature review

In this section, we provide a summary of the literature on two aspects: (1) research on two-sided markets, and (2) the impacts of service quality on decision making. The first aspect examines the relationship between previous research on two-sided markets and our own work. The second aspect reviews existing studies on service quality, some of which support our key assumptions and model development.

2.1 Researches on two-sided market

The popularity of two-sided markets in real life has drawn the attention of scholars to the related issues in this field. Instead of directly providing products or services, platforms act as“intermediary”connecting both sides of the market. The on-demand service platform, which is the focus of this paper, is just one example of such platforms, with similar forms found in rental markets(Benjaafar et al.,2015), software development markets, and so on. The primary research areas in this field include the pricing behavior of platform owners(Li et al.,2016), the matching mechanisms employed by the platform(Hu and Zhou,2017), and the effects of new platforms entering the market on existing ones(Zervas et al.,2016).

One prominent characteristic of two-sided markets is the presence of cross-network externalities. This means that the participation of service providers on the platform can impact the utility of demanders, thus influencing their decision to participate. In a two-sided market, the benefits of one party are closely tied to the scale of participation by the other party on the platform(Armstrong,2006). Users on both sides are considered valuable internal resources for the platform, and the initial user base is crucial for maintaining a competitive advantage and influencing long-term competition(Sun and Edison,2009).In a two-sided market, users on both sides derive utility or income by engaging in transactions on the platform. The impact of network externalities on pricing in two-sided markets has been studied extensively, including cross-group network externalities and intra-group network externalities(Rysman,2009;Bernstein et al.,2020). Many scholars argue that two-sided platforms exhibit the typical characteristics of cross-network externalities. This paper recognizes this feature and incorporates it into the utility function of service demanders.

The popularity of on-demand ride service platforms has attracted significant attention from scholars studying pricing decisions in this context. Some papers focus on the issue of price and wage incentives to effectively coordinate the supply and demand on these platforms(Eisenmann et al.,2014;Benjaafar et al.,2021;Chen and Hu,2020). Guda and Subramanian(2017)have studied the impact of surging price on driver positioning in a two-stage model. They find that drivers may not trust the demand prediction, so price changes can increase the credibility of the prediction. However, increasing prices for high-demand location-based areas may not always be the most effective strategy, as it can suppress demand growth or even drive drivers away from those areas.

Similarly, there have been matching studies that show how platform incentives for agents can influence the number of agents(Cachon,2003). In this paper, the optimal decision of the platform is determined by considering the service quality of drivers and balancing the supply and demand. The model is built upon the assumptions of previous studies.

2.2 Service quality on decision making

Currently, research on the service quality of platforms is mostly focused on the impact of waiting time during travel, as long waiting times often reduce consumers’service experience(Wang et al.,2007). Considering consumers’sensitivity or impatience towards waiting time, it is important to develop price and wage strategies to maximize profits under the assumption that consumers have low tolerance for waiting time(Bai and Tang,2018). Some consider how to enhance the matching efficiency of platforms. They thought platforms can improve service quality in the following aspects. Zhou et al.(2014) investigated the optimal pricing decisions of service enterprises with the presence of two groups of consumers having different valuation and service sensitivity. Ni et al.(2013) studied the optimal pricing and service speed of a platform with two types of consumers. The study found that under the goal of maximizing profits, it is not always optimal to serve all consumers. The waiting time acts as a determinant of demand, indicating the impact of service quality(Zhao et al.,2020).

In fact, consumers have different preferences and priorities, so differences in service quality need to be considered in some problems(Zhong et al.,2020). The preferences of consumers for platform service quality play a crucial role in determining the platform’s optimal decision strategy(Ji and Wang,2014). And service network externalities are often considered as part of service quality(Lu et al.,2018;Gui et al.,2021).This paper incorporates the heterogeneity of consumers’ service preferences in our model. Similar to the problem in the paper of Zhong et al.(2019), we studied the influence of each parameter on the optimal decision of the platform from the perspective of distinguishing and not distinguishing service quality. Similar to the problem addressed in Zhong et al.(2019)’s paper, our study examines the impact of various parameters on the optimal decision-making of the platform, focusing on the distinction and non-distinction of service quality. While they concluded that serving all consumers is not always optimal, our article takes a different approach by defining service quality as the consumer’s sensitivity to congestion levels. The consumer’s utility includes intrinsic evaluation, price, and waiting time, which serves as a proxy for service quality. By comparing the results, they illustrate the conditions under which differentiated or undifferentiated services are applicable and conclude that blindly serving all consumers is not the optimal strategy for the platform. Building upon these findings, our paper extends the concept of service quality by incorporating the consumer’s experience as a measure, in addition to considering waiting time. Furthermore, we also take into account the cross-network externalities. Recognizing the heterogeneity in service quality among consumers, we explore the service provider’s role and examine the optimal decision-making process while considering the service quality provided by the driver as parameters in our model.

Existing literature on the service quality of on-demand ride platforms mainly focuses on exploring the impact of service differences on both sides of the platform, or considers service preference as fixed parameter. However, considering the characteristics of cross-network externalities in a two-sided market, this paper takes into account the service quality types of drivers and the heterogeneous preferences of consumers for driver service quality. Our study aims to investigate the optimal pricing strategy for the platform in light of these factors.

Considering all the above research on on-demand service platform, we can fill the following gaps in literature:

(1)In this paper, the service provider's service quality is taken into consideration. Combined with the feature of two-sided market: cross network externalities, we considers the consumer's heterogeneity preference, service quality to explore how consumers integrated all factors to make decisions. Backward induction method is used to determine the platform optimization strategy.

(2)This study takes into account the service quality of the service providers, as well as the cross network externalities inherent in a two-sided market. By considering the heterogeneity in consumer preferences and the impact of service quality on their decision-making process, we aim to understand how consumers integrate all these factors to make their choices.

 In the context of this study, the platform initially enters the market by offering a single type of service without differentiating between providers. All providers pool together to offer this service to consumers. The platform’s decision variable is a single set of price and wage. This paper is the first to compare platform decisions with different parameters and identify the conditions under which the platform should provide a single service or opt for differentiated services. By analyzing the impact of various factors, such as provider service levels and retention costs, the study determines the optimal strategy for the platform to maximize its profits while satisfying consumer preferences.

Overall, by addressing these gaps, our research contributes to a more comprehensive understanding of service quality and pricing strategies in on-demand ride platforms. We provide valuable insights for platform decision-makers to optimize their strategies and enhance the overall service quality and customer satisfaction.

Finally, it’s very kind of you to provide us so many comments and suggestions to improve this paper.

Thanks a lot!

---

## [Decision Letter · Decision Letter 1]

19 Dec 2023

On-demand ride service platform with differentiated services

PONE-D-23-03432R1

Dear Dr. Wei,

We’re pleased to inform you that your manuscript has been judged scientifically suitable for publication and will be formally accepted for publication once it meets all outstanding technical requirements.

Kind regards,

Juan Carlos Rocha Gordo

Academic Editor

PLOS ONE

Additional Editor Comments (optional):

Dear authors,

Apologies for a longer than normal waiting time with your manuscript. Two reviewers have now assessed your work and agree that the manuscript is ready for publication. The reason for long review time is that I really wanted to secure a second reviewer for your paper. On the first round only one reviewer assessed the work, and I acted as second reviewer. However, your research is not on my area of expertise and I wanted to secure an impartial view on it. I've invited over 15 additional reviewers, all of them declined, and many others did not respond. That is why the review time took so long.

Best regards,

Juan Rocha

Reviewers' comments:

Reviewer's Responses to Questions

**Comments to the Author**

1. If the authors have adequately addressed your comments raised in a previous round of review and you feel that this manuscript is now acceptable for publication, you may indicate that here to bypass the “Comments to the Author” section, enter your conflict of interest statement in the “Confidential to Editor” section, and submit your "Accept" recommendation.

Reviewer #1: All comments have been addressed

Reviewer #2: All comments have been addressed

2. Is the manuscript technically sound, and do the data support the conclusions?

Reviewer #1: Yes

Reviewer #2: Yes

3. Has the statistical analysis been performed appropriately and rigorously? 

Reviewer #1: N/A

Reviewer #2: I Don't Know

4. Have the authors made all data underlying the findings in their manuscript fully available?

Reviewer #1: Yes

Reviewer #2: Yes

5. Is the manuscript presented in an intelligible fashion and written in standard English?

Reviewer #1: Yes

Reviewer #2: Yes

6. Review Comments to the Author

Reviewer #1: I appreciate the author's revisions. The manuscript has been improved and issues addressed, showing progress. It now essentially meets the publication standards. While there's room for further refinement, it presents a commendable effort worthy of consideration for publication.

Reviewer #2: (No Response)

7. PLOS authors have the option to publish the peer review history of their article (what does this mean?). If published, this will include your full peer review and any attached files.

Reviewer #1: No

Reviewer #2: No

---

## [Editor Report · Acceptance letter]

3 Jan 2024

PONE-D-23-03432R1 

PLOS ONE

Dear Dr. Wei, 

I'm pleased to inform you that your manuscript has been deemed suitable for publication in PLOS ONE. Congratulations! Your manuscript is now being handed over to our production team.

Kind regards, 

on behalf of

Dr. Juan Carlos Rocha Gordo 

Academic Editor

PLOS ONE